## Registered report

behaviour

action copying, chimpanzees, novelty, rearing background, anchored actions, environmental effects

**Author for correspondence:**
Alba Motes-Rodrigo
e-mail: albamotes7@gmail.com

# Evaluating the influence of action- and subject-specific factors on chimpanzee action copying

Alba Motes-Rodrigo[1], Roger Mundry[2,3], Josep Call[4] and Claudio Tennie[1]

[1]Department of Early Prehistory and Quaternary Ecology, University of Tübingen, Tübingen, Germany
[2]Max Planck Institute for Evolutionary Anthropology, Leipzig, Germany
[3]Platform Bioinformatics and Biostatistics, VetMedUni Vienna, Vienna, Austria
[4]School of Psychology and Neuroscience, University of St Andrews, St Andrews, UK

AM-R, 0000-0002-4444-7723; CT, 0000-0002-5302-4925

The ability to imitate has been deemed crucial for the emergence of human culture. Although non-human animals also possess culture, the acquisition mechanisms underlying behavioural variation between populations in other species is still under debate. It is especially controversial whether great apes can spontaneously imitate. Action- and subject-specific factors have been suggested to influence the likelihood of an action to be imitated. However, few studies have jointly tested these hypotheses. Just one study to date has reported spontaneous imitation in chimpanzees (Persson *et al.* 2017 *Primates* **59**, 19–29), although important methodological limitations were not accounted for. Here, we present a study in which we (i) replicate the above-mentioned study addressing their limitations in an observational study of human–chimpanzee imitation; and (ii) aim to test the influence of action- and subject-specific factors on action copying in chimpanzees by providing human demonstrations of multiple actions to chimpanzees of varying rearing background. To properly address our second aim, we conducted a preparatory power analysis using simulated data. Contrary to Persson *et al.*'s study, we found extremely low rates of spontaneous chimpanzee imitation and we did not find enough cases of action matching to be able to apply our planned model with sufficient statistical power. We discuss possible factors explaining the lack of observed action matching in our experiments compared with previous studies.

# 1. Introduction

The ability to imitate, understood as copying the form of an action [1], especially of cognitively opaque behaviours [2,3], has been deemed crucial for the emergence of human forms of culture [4]. However, the ability for spontaneous imitation in non-human great apes is still a controversial topic [5].

Different studies on *enculturated* apes [6] aimed at directly testing whether apes could imitate actions demonstrated to them. Tomasello *et al.* [2] tested three enculturated apes (a 10-year-old bonobo, a 5-year-old bonobo and a 5-year-old chimpanzee) and three unenculturated apes (a 21-year-old bonobo, a 3-year-old bonobo and a 7-year-old chimpanzee) on their ability to imitate human-demonstrated actions on novel objects. A baseline of 4 min with each of the novel objects was included for each individual. The authors found that mother-reared apes were much poorer imitators than enculturated apes when object-related actions were demonstrated [2]. Other studies have used the 'Do as I Do' paradigm to study imitation in apes. In this paradigm, subjects are trained to imitate demonstrated familiar actions and then continue to imitate when tested with novel actions. Crucially, this paradigm has been shown to elicit changes in white matter connectivity in great apes [7]. Custance *et al.* [8] trained two juvenile chimpanzees with 15 taught actions in a 'Do as I Do' paradigm for three-and-a-half months. Later, the authors demonstrated 48 actions to the two chimpanzees, which the authors considered to be novel for the subjects. The two chimpanzees reportedly copied 13 and 17 of these actions respectively. In a later study, Hribar *et al.* [9] trained one nursery-reared chimpanzee (6 years old) in the 'Do as I Do' paradigm for seven months with nine familiar actions using clicker-training and 15 novel actions (also used by Custance *et al.* [8]) through shaping. After 92 sessions, the chimpanzee reportedly correctly reproduced 80% of the demonstrated actions. When the chimpanzee was presented with 45 *novel* actions not included in the training, he was reported to correctly reproduce 27% of them [9].

Call [10] tested one enculturated orangutan on his ability to imitate the same demonstrated actions previously used by Custance *et al.* [8]. The orangutan showed different accuracies of action matching, depending on whether the body part involved in the action was gross (e.g. head, arms) or detailed and whether the action involved contact with his own body. Byrne & Tanner [11] opportunistically presented an enculturated gorilla with seven actions not considered to be species-typical. Action matching of varying accuracy was detected for five out of the seven demonstrated actions. However, those actions that were classified as closely matching the demonstrated actions were actually found not to have been novel to the gorilla (as they had previously been observed in the same subject on film records spanning 5.5 years).

As exemplified by the study conducted by Byrne & Tanner [11], action imitation studies interested in novel action copying present an important caveat: it is problematic to determine whether the demonstrated behaviours are truly novel. It has been suggested that the active recollection of actions already present in the behavioural repertoire (contextual imitation) as a result of seeing a demonstration (i.e. via response facilitation) represents a different phenomenon from the imitation of actions that are not already present in the behavioural repertoire, which forms need to be acquired for the first time (production imitation; [12]).

Three studies have tested *unenculturated* apes on their ability to imitate actions produced by human demonstrators. Tomasello *et al.* [13] separately trained two female chimpanzees (ages 27 and 22 years) from two different groups to perform one and two gestures, respectively, in order to obtain food from a human. After reliably performing the trained gesture, the two females were returned to their groups where they performed the trained gestures while other individuals were present. The actions included in the trained gestures ('raise arms above head', 'run hands along fence', 'place head on fence') were not completely novel to the trained chimpanzees, and it is not specified whether the other chimpanzees were unfamiliar with these actions. None of the subjects that observed the two chimpanzee demonstrators produced the target gestures [13]. Tennie *et al.* [14] expanded and addressed some of the limitations discussed in Tomasello *et al.* [13] by training one male chimpanzee to demonstrate familiar and unfamiliar actions in begging and non-begging contexts. Baselines, in which no demonstrations took place, were used in the different experiments. Only one male chimpanzee performed a familiar demonstrated action in a begging context, which was interpreted as evidence of contextual imitation [14]. No imitation of unfamiliar (novel) actions took place [14]. Finally, Clay & Tennie [15] used a $2 \times 2$ design in which they demonstrated familiar, unfamiliar, anchored (involving contact) and intransitive actions to 46 bonobos. These actions were performed before accessing a reward placed inside the object to which the actions were directed. No bonobo copied any of the demonstrated anchored or intransitive actions [15]. Therefore, there is

no evidence suggesting that unencultured apes perform production imitation, and there exists only very limited evidence that such apes perform contextual imitation [11,14].

To the best of our knowledge, only one study to date [16] has claimed to present data showing *widespread* and *untrained* imitation in non-human great apes. Persson *et al.*'s study focused on the interactions between five chimpanzees (four partly human-reared and one mother-reared) at a European zoo and human visitors. The authors studied these interactions both when the visitors were indoors (close to the chimpanzees) and outdoors (when visitors were approx. 6 m away from the chimpanzees). The authors coded for accuracy of the action match and the species that initiated the imitative episode (i.e. human or ape demonstrator). In stark contrast to the studies presented above, the authors claimed that the relative frequency of action copying was similar in the two species: '[The two] species imitated each other at a similar rate corresponding to almost 10% of all produced actions' though they also reported that 'exact imitation (as opposed to partial imitation) was observed significantly more often ($Z = 2.93$, $p = 0.003$) in humans (73% of all imitative actions, $n = 151$) than in chimpanzees' [16].

How could these data fit in with the data previously collected on ape imitation? On the one hand, it could be that previous studies have failed to detect spontaneous ape imitation. On the other hand, the discrepancy between previous studies and the study of Persson *et al.* [16] could be due to confounds and limitations neglected by the latter. First, Persson *et al.* did not include a baseline (repertoire of behaviours that the apes perform without being under any experimental condition). Second, they did not collect video recordings of the human–chimpanzee interactions, which would have allowed for independent behavioural coding (as a check for reliability of the observations). Third, the possibility remains that the chimpanzees never really copied the humans, and that it only had appeared so for the human observers of the human–ape interactions (e.g. some cases when the humans imitated the apes might have appeared as the other way around). Fourth, the criteria for determining if an action was imitative were that it should not have been observed in the same episode or in the previous 3 min [16]. However, this criterion does not exclude the possibility that the actions were present in the behavioural repertoires of the chimpanzees but just not performed at the precise moment that observations took place. Fifth, the potential imitative abilities shown by the chimpanzees in Persson *et al.* might have arisen from enculturation effects of human-rearing.

The first aim of our study is to test if the results presented by Persson *et al.* can be replicated when the potential limitations and confounds listed above are controlled for. Contrary to Persson *et al.* [16], our study includes a behavioural baseline and video recordings of the interactions between humans and apes, which allows to differentiate between contextual and production imitation as well as to assess inter-observer reliability (Visitor experiment). The second aim of our study is to jointly test if the different factors suggested to influence imitation in apes, contribute to eliciting imitation of human-demonstrated actions in chimpanzees (Demonstration experiment).

The specific aims of the Visitor experiment are (i) to assess if chimpanzees and/or zoo visitors spontaneously imitate any actions of the other species and (ii) to test if human biases exist in perceiving that the chimpanzees imitate visitors even when action matching does not actually take place (as a potential alternative explanation of the findings of [16]). Our null hypotheses for this experiment are that neither chimpanzees nor zoo visitors spontaneously imitate each other (aim (i)) and that humans are not biased towards perceiving imitation (aim (ii)). The Demonstration experiment will assess if chimpanzees imitate actions performed by human demonstrators when tested individually. If enough evidence for imitation of demonstrated actions is found, we will assess (i) if the rearing background influences chimpanzees' imitative abilities (following [2]), (ii) if chimpanzees engage in contextual imitation of familiar actions, production imitation of unfamiliar actions or both (following [11]), (iii) if actions are imitated depending on the actions being 'anchored' or not (i.e. actions involving contact with the subjects body, following [10]), and finally (iv) if actions are imitated depending on the presence of an environmental effect (sound) or not—as sound reproduction can be done via emulation instead of imitation (following [13]). Our null hypotheses for this experiment are that chimpanzees do not imitate actions performed by human demonstrators and that neither the rearing background (aim (i)), the familiarity of the action (aim (ii)), action anchoring (aim (iii)) nor the presence of environmental effects (aim (iv)) influence the probability of imitation. As it is only possible to test these hypotheses if sufficient statistical power is achieved, we have conducted a series of power analyses to determine the number of imitation events we would need to observe in the Demonstration experiment to test these hypotheses and the probability of being able to apply the corresponding statistical models.

# 2. Methods

## 2.1. Study site and subjects

We studied the chimpanzee group housed in Leintal Zoo in Schwaigern, Germany, which consists of 32 chimpanzees ($N_{males}$ = 15, $N_{females}$ = 17) ranging in age from 5 to 47 years (at the time of writing, mean ± s.d. = 24.28 ± 9.25 years). All chimpanzees are mother-reared except for seven chimpanzees who were human-reared by the zoo staff. The chimpanzees have access to an indoor (282.8 m$^2$) and an outdoor enclosure, as well as seven sleeping rooms (47.3 m$^2$ together). The outdoor enclosure is composed of four interconnected areas, which together add to 958.25 m$^2$. These areas are surrounded by a cubic metallic mesh 4.7 m high that limits the enclosure sideways and from the top, acting as a roof. The outdoor areas are connected via several manual sliding doors and two doors that connect with the indoor enclosure. The doors connecting to the indoor enclosure are kept open during the day. The sleeping rooms are out of sight from the visitors and are kept open during the day except during cleaning hours. Structural enrichment consisting of climbing structures, logs, ropes, platforms and hammocks is present both in the indoor and outdoor enclosures. The main diet of the chimpanzees is composed of fruits and vegetables with occasional bread (the latter approximately once a week). Peeled seeds and oats are spread every morning in the outdoor enclosure, and honey is placed in the cavities of the climbing logs present in the outdoor enclosure. Occasionally (approximately once a week), rug sacks are provided to the chimpanzees as nesting material. Five public feedings are conducted every day in the outdoor enclosures and water is available ad libitum in the indoor enclosure.

## 2.2. Experimental set-up

Two experiments (Visitor experiment and Demonstration experiment) were conducted in the present study between the 1st and the 23rd August 2020. Each experiment was analysed independently. No between-experiment comparisons were conducted due to the differences in set-up resulting from the different aims of each experiment.[1]

Prior to the experiments, we compiled a behavioural baseline of all the behavioural forms present in the chimpanzee population where the tests took place (electronic supplementary material, Ethogram). This action repertoire was compiled at group level,[2] without taking into account the subject's identity, as the aim of the baseline is to determine which behavioural forms the chimpanzees can be expected to show/know prior to the main study. The action repertoire consists of both behaviours directed towards other chimpanzees and behaviours directed towards humans (visitors and keepers). Video recordings (38 h) of social interactions among chimpanzees and between chimpanzees and humans (visitors and keepers) were collected from 9.00 until 15.00 during 10 days in October 2018. Videos were collected sporadically at different times of the day in order to account for varying visitor numbers. As an example, recordings were made both during public feedings (when the number of visitors varied between 10 and 30) and in the last opening hour of the day when most visitors had already left. Recordings took place close to the outdoor enclosures of the chimpanzees with a Sony Handycam HDR-CX450.

During the Visitor experiment, 50 zoo visitors above 18 years old were recruited while they were in the zoo using the attached forms in the electronic supplementary material, S1 'Declaration of consent', 'Declaration of consent for video and audio recordings' and 'General information for visitors'. If participants signed these consent forms, we additionally collected some demographic information of the participants via the 'Questionnaire about participants' information'. We used regular zoo visitors in this experiment to maintain an experimental set-up as similar as possible to the study of Persson *et al.* [16]. The participants were recruited before they reached the chimpanzee enclosure and recordings (via two cameras on tripods) started as soon as the visitors established visual contact with the chimpanzees in order to ensure that we captured on video all chimpanzee behaviours that the

---

[1]Comparisons between our two experiments did *not* take place because each experiment was designed with a different set-up in order to test experiment-specific aims. A between-experiment comparison was not conducted, as confounders such as the familiarity of the demonstrators, the location of the test (indoors and outdoors) and the set-up (individual versus group testing) would have made the comparison uninterpretable.

[2]In this study, we did not determine the identity or the number of individuals that were observed performing each of the behaviours included in the baseline. Our main goal when compiling the baseline was to record all behaviours present in the population regardless of the identity or the number of chimpanzees that perform each individual behaviour.

**Table 1.** Actions demonstrated to the chimpanzees during the Demonstration experiment. When the action involves only one hand, the hand used (left or right) was not taken into account.

| familiar action | | | non-familiar action | | |
|---|---|---|---|---|---|
| | environmental effect | non-environmental effect | | environmental effect | non-environmental effect |
| contact | tap surface (bars, floor or ceiling—everything else will have been removed) with finger/fingers of one hand making a sound | touch own shoulder with finger/fingers/ palm slowly (without producing a clapping sound) | contact | strum lips (to repeatedly strum the upper and lower lips using one or several fingers of one hand producing a sound) | Place fingers in front of lips with the index finger side touching the lips. All fingertips are kept upright and the remaining fingers are away from the face (like a silencing gesture but without making a fist). |
| non-contact | — | swing both arms (below shoulder level) 'in the air' | non-contact | — | praying hands (see [14]) |

visitors saw. This procedure allowed us to assess whether the visitors demonstrated actions to the chimpanzees that they had previously seen the chimpanzees perform (human visitors imitating chimpanzees) or not. Participants were only considered for the study if they had never visited the chimpanzee enclosure before in order to prevent carry-over effects from actions observed in previous visits. If the visitors had already been at the chimpanzee enclosure where tests took place or did not sign the consent forms, they were excluded from the experiment. The experimenter was present during the trials and supervised the interactions. Each participant in the experiment was filmed in a delimited area at a glass window of the enclosure where no other visitors were present[3] (in order to not film any visitor that had not provided written consent). The participants in this experiment received another questionnaire when they stopped engaging with the chimpanzees (which they could do any time) to evaluate their interactions (see the 'Final questionnaire' in the electronic supplementary material, S1 for details on the questions). The visitors were free to finalize the interactions with the chimpanzees at any time they wanted and they were not asked to interact for any minimum amount of time. Data collection stopped when 50 first-time zoo visitors had participated in the experiment.

During the Demonstration experiment, a keeper working at the zoo who was familiar with the chimpanzees performed a series of previously selected actions to the chimpanzees in their sleeping quarters. Although most tests were performed by the same keeper, four chimpanzees were tested by other keepers due to working shifts. Tests were conducted individually (one chimpanzee at a time) following Call [10] and Byrne & Tanner [11]. This experiment used a $2 \times 2 \times 2$ design in which demonstrators performed familiar (present in the repertoire compiled in baseline condition) and non-familiar actions (likely to be absent from the chimpanzees' gesture repertoire and checked for complete absence in our baseline) to both mother- and human-reared chimpanzees. The demonstrations included actions with and without environmental effect (sound) as well as anchored (contact with the subject's body) and non-anchored (non-contact with the subject's body) actions (table 1).

---

[3]If the participants had performed a behaviour or action that could have been stressful/dangerous/scary for the apes (e.g. loud glass banging, screaming, throwing items, any attempt at provisioning), the experimenter would have immediately intervened and terminated the trial.

In total, six actions were demonstrated to each chimpanzee, counterbalanced across subjects. Non-contact actions with environmental effect were not included in the study as it would have required the use of remotely activated devices. Each demonstrated action was considered a trial and the chimpanzees' participation was voluntary. The testing institution houses a total of seven human-reared chimpanzees (one male chimpanzee died since the registration of this manuscript; 2M and 5F, Mean age = 27.2), all of which were included in the study. These individuals have in common that they have experienced extensive human contact during their first year of life. Some of the human-reared chimpanzees lived for a certain period of time in human homes (the exact length is not possible to determine as many of the individuals were abandoned at the zoo entrance by the previous private owners), whereas others lived in a nursery group of conspecifics at the zoo and were bottle fed every day because their mothers rejected them at birth. Given their extensive human exposure, these seven individuals can be considered to be enculturated to some degree [6]. In addition to the seven human-reared individuals, we tested the same number of mother-reared individuals ($N = 7$; 1M and 6F, Mean age = 19.1). We conducted a total of 84 trials (14 individuals tested in 6 trials each).

Each trial took place as follows: two cameras were placed on tripods filming the demonstrator and the chimpanzees. The demonstrator started each session by facing the chimpanzee being tested. Once the chimpanzee oriented its head towards the demonstrator, the first action was demonstrated (as in, e.g. [10]). The actions were demonstrated four times each with at least 10 s between demonstrations (each time waiting for the chimpanzee to have its head oriented towards the demonstrator). Each demonstration itself lasted approximately 10 s. After four demonstrations took place, the demonstrator waited for 20 s before moving on to the next action. In total, 336 demonstrations (including all six actions) were performed (four demonstrations in each of the 84 trials).

## 2.3. Data collection

To compile the ethogram that constituted our behavioural baseline (electronic supplementary material, Ethogram), we coded the gestural forms described in wild chimpanzees by Hobaiter & Byrne [17]. As the focus of the baseline is not to assess the use of intentional gestures but to assess the presence or absence of behavioural forms, no 'signs of intentionality' (audience checking, response waiting, persistence; [17]) were coded/required to classify a behavioural form as being present in the chimpanzees' repertoire. In addition, we also coded behavioural forms found in the study group that were absent in the study by Hobaiter & Byrne [17]. Before the onset of the experiments, we further asked the zookeepers individually if there were any other additional behaviours present in the chimpanzee population that we had not included in our baseline.

During the Visitor experiment, the experimenter live coded the following variables from each potential imitative event (for the purpose of this study, a potential imitative event was considered when the visitor was within 2 m of a chimpanzee through the glass window or the mesh of the outdoor enclosure).[4]

— Presence of action matching (yes or no): the chimpanzees or the humans perform an action within 5 s after an action has been demonstrated by an individual of the other species.
— Who initiated the imitative event according to the temporal order in which humans and chimpanzees performed the action: the first species observed by the experimenter during a participant's trial performing an action was coded as the initiator. (Note that this is hard to do via live-coding and could, therefore, represent one of the potential limitations of Persson *et al.* [16].)
— If the demonstrated action was anchored (involved contact with the subjects body or an object) or not.
— If the demonstrated action had an environmental effect (e.g. creates a sound) or not.
— If the demonstrated action was in the chimpanzees' repertoire compiled during the baseline condition (familiar action) or not.
— Time of day.

From the video recordings compiled during the visitor experiment, each potential imitative event was coded again by a second coder for the presence of action matching (these included both events when

---

[4]Note that imitative events can also take place at larger distances, but for the purpose of this study, the term 'potential imitative events' will be used when the distance between a chimpanzee and a zoo visitor is less than 2 m. This method was adopted based on the result of Persson *et al.* [16] that 50% of the actions presumably imitated by the visitors and 72% of the actions presumably imitated by the chimpanzees occurred at a distance smaller than 2 m.

the experimenter coded imitation and events when the experimenter did not code imitation). The initiating species in each event was also recoded from the video recordings by a second coder. We coded the potential imitative events a second time by an independent coder in order to determine if the experimenter was biased regarding the perception of imitation and the species that initiated imitative events. We also recorded a timeline of the potentially imitated actions across the trial of each participant to determine who showed each action first (human or ape) in order to differentiate between human and chimpanzee imitation. In addition, the following variables were coded from video recordings for each potential imitative event:

— Imitation time frame (short or long): the imitated action was performed within 1 min (short) or after 1 min (long) of the demonstrated action.
— The identity of the chimpanzee involved in the potential imitative event.
— The rearing history of the chimpanzee involved in the interaction (human- or mother-reared).
— The accuracy of the action match, defined as how closely the matched action resembled the demonstrated action. Action matching was coded as follows (adapted from [10]):
    (i) Full reproduction: the behaviour was fully reproduced.
    (ii) Partial reproduction: Some elements of the behaviour demonstrated such as the action or the body parts involved were missing. Two subcategories can be distinguished depending on the missing elements:
    *Body part*: The action was correct but the body part that executed it was incorrect.
    *Action*: The body parts that executed the action were correct, but the performed action was incorrect.
    (iii) Failed reproduction: the subject performed an action completely unrelated to the one demonstrated by the model.
— The duration of the visitor engagement, defined as the time the visitor spends within 2 m of the enclosure windows facing the chimpanzees.

In order to conclude that imitation took place, action matching must have occurred and the reproduction had to be either full or at least partial.

In addition, information regarding the perceptions of the visitors regarding if the chimpanzees imitated them was collected via the 'Final questionnaire'.

During the Demonstration experiment, the following variables were coded from video recording from each action demonstration:

— Presence of action matching (yes or no): the chimpanzees performed an action after an action was demonstrated.
— Imitation time frame (short or long): the imitated action was performed within 1 min (short) or after 1 min (long) of the demonstrated action.
— Demonstration number.
— The identity of the chimpanzee.
— The rearing history of the chimpanzee (human- or mother-reared).
— If the demonstrated action was anchored or not.
— If the demonstrated action had an environmental effect (sound) or not.
— If the demonstrated action was in the chimpanzees' repertoire compiled during the baseline condition (familiar action) or not.
— The accuracy of the action match.

Prior to the start of testing, the human demonstrator was instructed not to imitate any action the chimpanzee performs during the trials. In order to conclude that imitation took place, action matching must have occurred and the reproduction had to be either full or at least partial.

## 2.4. Data analysis

The aims of the Visitor experiment are to determine (using video recordings as well as live coding) if there exists interspecies imitation between humans and chimpanzees as well as to test for potential human biases in perceiving imitation regarding the species that initiates the imitative events (e.g. when those who copied the action are the visitors instead of the chimpanzees).

To address the first aim of this experiment, the presence or absence of imitation was coded from potential imitative events both live by the experimenter and from video recordings by a second coder.

**Table 2.** Description of the variables included in the models.

| element | description | variable type | levels or range |
|---------|-------------|---------------|-----------------|
| response | presence or absence of imitation | binomial | 2 (0 or 1) |
| test predictor | action type | factor | 6 |
| | rearing | factor | 2 (mother- or human-reared) |
| control predictor | age | covariate | 5–47 |
| | age-squared | covariate | |
| | demonstration number | covariate | 1–4 |
| | trial number | covariate | 1–6 |
| | interaction between trial and demonstration number | interaction | |
| random effects | subject | factor | 16 |
| random effect structure | random intercept and random slopes of action (dummy coded and centred), trial number, demonstration number and their interaction within subjects | | |

Each potential imitative event identified during live coding was entered as an independent data point. A second assessment of the true presence (action matching within 1 min of a demonstration, [10]) or absence of imitation of live-coded events was established via video recordings by the second coder. We increased the time delay (from 5 s to 1 min) when coding videos compared with live coding, due to the difficulty of determining/detecting long delays during live coding. For each imitative event in the dataset, we coded if it was detected via live coding, video recordings, or both. If the two coders did not agree on the presence or absence of imitation, the event was labelled as ambiguous. As a different variable, we coded if the visitor that participated in a given trial *perceived* that the chimpanzees imitated her at some point during the trial (see 'Final questionnaire' in the electronic supplementary material, S1).

From the timelines created in each trial of this experiment, we coded which species performed an action for the first time during a trial in order to determine if the actions demonstrated by the visitors had been previously performed by the chimpanzees during the trial (humans copying chimpanzees) or not (chimpanzees copying humans). The second coder, therefore, coded the presence or absence of imitation (action matching), which species initiated the imitative event, and the accuracy of the match.

The inter-observer reliability between the two assessments of the potential imitative events (from live coding and video recording) was used to address the second aim of this experiment regarding the presence or absence of human biases in the perception of imitation using the Cohen's $\kappa$ statistic.

The aim of the Demonstration experiment is to systematically determine if and what actions do chimpanzees spontaneously imitate from human demonstrators. If imitation does occur, the additional aim of this experiment is to determine which factors influence its appearance. To address the first aim, the experimenter and a second coder coded all trials for the presence or absence of action matching and for the accuracy of the match. In those cases where the two coders did not agree in the presence or absence of imitation, the trials were excluded from further analysis. Inter-observer reliability was established using Cohen's $\kappa$ statistics.

In order to determine which action- and subject-specific factors elicit imitation, it is necessary to possess a sufficient number of imitation events in order to fit a model (table 2) with sufficient power to investigate this question. To evaluate the power that our analysis would have, we *a priori* simulated 1000 datasets in which the probability of imitation was influenced by the type of action demonstrated (i.e. the particular combination of familiarity, presence of environmental effect and contact involvement; table 3) and by the chimpanzee's rearing history. Based on the literature, we assumed that mother-reared chimpanzees have half (a conservative estimate) the probability of imitating an action of human-reared chimpanzees [2]. We did not expect imitation probability to be affected by age, trial number or demonstration number, so we did not simulate the respective effects of these variables (but still accounted for them in the models fitted to the simulated data; see below;

**Table 3.** Simulated relative probabilities of each action type to be imitated relative to the maximum probability of imitation, which is assumed to be that of familiar contact actions with environmental effects. Relative probabilities were estimated based on previous studies.

| | familiar action | | non-familiar action | |
| --- | --- | --- | --- | --- |
| | environmental effect | non-environmental effect | environmental effect | non-environmental effect |
| contact | Pmax | $0.8 \times$ Pmax | $0.7 \times$ Pmax | $0.5 \times$ Pmax |
| non-contact | — | $0.4 \times$ Pmax | — | $0.1 \times$ Pmax |

table 2). In the simulated datasets, we varied the overall probability of imitation, fitted several models to each of the simulated datasets, and then determined for each of them whether it converged as well as the significance of the full–null model comparison (see below) and of the individual terms in the model.

We simulated the predictors assuming a total of 16 individuals, each tested with all six actions, each demonstrated four times per trial, leading to a total sample size of 384 observations per dataset. Although we aimed to test all 16 individuals included in the simulation, one chimpanzee deceased shortly before data collection began and thus (to keep the number the number of human- and mother-reared individuals equal), seven chimpanzees from each rearing background were tested (total $N = 14$). Each individual in the simulation got assigned one value for whether it was mother-reared (randomly sampled with equal probabilities to be 'no' or 'yes') and an age randomly sampled from a uniform distribution with a minimum of 5 and a maximum of 47.

We assigned each action a different probability of being imitated, relative to Pmax, the maximum probability of imitation (table 3). Importantly, these simulated relative probabilities do not reflect our expectations, which are less specific, given the very scarce data existing in this field. In fact, if actions are imitated, we expect that their probabilities will vary among the cells of the design in a manner potentially equivalent to a three-way interaction between the action-specific factors. Hence, the simulated probabilities only mirror our expectations regarding the broad pattern of imitation probabilities in the sense that they might depend on the particular combination of familiarity, contact involvement and environmental effect in a complex way. We simulated all other terms in the model with the exception of rearing (see above) to have no effect on the response.

We simulated 18 different maximum probabilities of an action to be imitated (Pmax set to 0.0001, 0.0005, 0.001, 0.005, 0.01, 0.02, 0.03, 0.04, 0.05, 0.06, 0.07, 0.08, 0.09, 0.1, 0.2, 0.4, 0.6 or 0.8).

We simulated 1000 datasets for each maximum probability and fitted several full and null models to each of them. The reason for fitting several full and null models was that we wanted to explore whether fitting simpler models would increase power. The models differed mainly in which fixed effects they included for modelling how imitation probabilities differed between types of actions (electronic supplementary material, table S1). The simplest model only included the three main effects of action familiarity, presence of environmental effect and contact involvement, whereas the most complex model included also all three two-way interactions between these factors. Three additional models each included all three main effects, but each of them also one two-way interaction between two of them. Finally, we fitted a further model, which addressed the effect of the three-way interaction between the three factors. Since we were not going to have observations in each of the eight combinations of levels of the three factors, the three-way interaction between them cannot be included in the model. Instead, we included a single factor 'action type', which labelled the particular combination of action familiarity, presence of environmental effect and contact involvement. This model cannot be compared with any of the simpler models by means of a significance test since the terms they include are not all present in the more complex model. Therefore, we compared this model with the simpler ones (and also the simpler ones with one another) by means of Akaike's information criterion, corrected for small samples [18]. Each model comprised a random slopes structure within individual [19,20] fully mirroring its fixed effects structure regarding how we modelled the effects of the type of action (whereby we manually dummy coded and then centred the factors) and also included interactions if present in the fixed effects part. We included in each model fixed effects for rearing, the age of the individuals (linear and squared) as well as trial number and demonstration number and their two-way interaction (for trial number, demonstration number and their interaction we also included random slopes within individual).

The null models lacked the factors modelling the type of action and rearing history (reference category set to mother-reared), and the comparison between the two allows for an overall test of the effects of these factors which controls for 'cryptic multiple testing' [21]. We conducted a full–null model comparison only for simulated datasets for which both the null and full model converged. The models were generalized linear mixed models (GLMM; [22]) with binomial error structure and logit link function [23], fitted in R (v. 3.6.1; R Core Team 2019) using the function glmer of the package lme4 (v. 1.1-21; [24]) in combination with the optimizer 'nloptwrap'. The covariates (age, trial number and demonstration number) were z-transformed before including them in the models. When the full model converged, we also determined the significance of each of the terms in the model by dropping them from the full model one at a time. We determined power as the proportion of datasets that revealed a significant full–null model comparison (i.e. considering datasets for which the full and/or the null model failed to converge as non-significant). If the simulated response consisted solely of zeros, we did not fit any model and considered this dataset as non-significant, too. We further evaluated, separately for the individual predictors, the proportion of datasets in which they revealed significance.

### 2.4.1. Results of the simulation

When the maximum imitation probability (Pmax) was low, the number of datasets in which the simulated response consisted entirely of zeros was fairly large, and hence the number of datasets for which we even attempted to fit a model was low ('nr. models attempted' in electronic supplementary material, figure S1). The number of models that converged without an error clearly increased with Pmax. For imitation probabilities between approximately 0.005 and 0.03, a considerable fraction of the models we attempted to be fit revealed an error. The probabilities to converge without an error of the different models fitted did not differ much between models, although for the model comprising all two-way interactions and the model including the two-way interaction between familiarity and environment, it was slightly lower than for the other ones (electronic supplementary material, figure S1).

The probability of a model to converge without any warnings was, in part, much lower than the probability of the models to converge without an error (compare electronic supplementary material, figures S1 and S2). In fact, only when Pmax was 0.4 or larger, did this probability exceed 0.8. At intermediate values of Pmax (ca 0.02–0.1), the different models differed to some extent with regard to their probability to converge, but at low and high values of Pmax, these differences were low. It should be noted here that non-convergence means that model coefficients and the assessment of significance are potentially very unreliable. However, it is possible to set the optimizer to 'bobyqa' rather than 'nloptwrap' (as used here), which has a higher chance to converge (the reason that we used the latter in our simulations is that it is usually faster). Hence, our assessment of convergence probability might be too pessimistic.

The power of the full–null model comparisons (i.e. the probability of a significant result) only reached levels of approximately 0.8 for maximum imitation probabilities being 0.4 or above and was considerably below 0.8 for all smaller maximum imitation probabilities (figure 1). The differences between different ways of parametrizing action type were rather low and most prominent for intermediate maximum imitation probabilities between approximately 0.04 and 0.2. When determining power not as the proportion of simulated datasets that revealed a significant full–null model comparison but as the proportion out of those datasets for which both the full and null model converged without any warning, the power was clearly increased (except for very low maximum imitation probabilities) and also for a maximum imitation probability of 0.2 power was approximately 0.8 (figure 2). As stated above, it seems plausible to assume that at least part of the models which did not converge would actually converge after changing the optimizer, so the true power probably lies in between that depicted in figures 1 and 2.

For the effect of rearing, we found that it revealed a high power for maximum imitation probabilities of being 0.6 or larger whereby the way we parametrized action type did not have much influence on the power achieved (figure 3). The power of detecting the main effects of action familiarity, contact involvement and presence of environmental effect were investigated for the model comprising only the main effects, and also for the three models comprising one two-way interaction each, for the predictor not involved in the respective interaction. We found that power was highest for contact involvement and lowest for the presence of environmental effect (figure 4). A power of approximately 0.8 or more was only reached for contact and familiarity and only when the maximum imitation probability was at least 0.6. The power for detecting a given main effect as significant did not

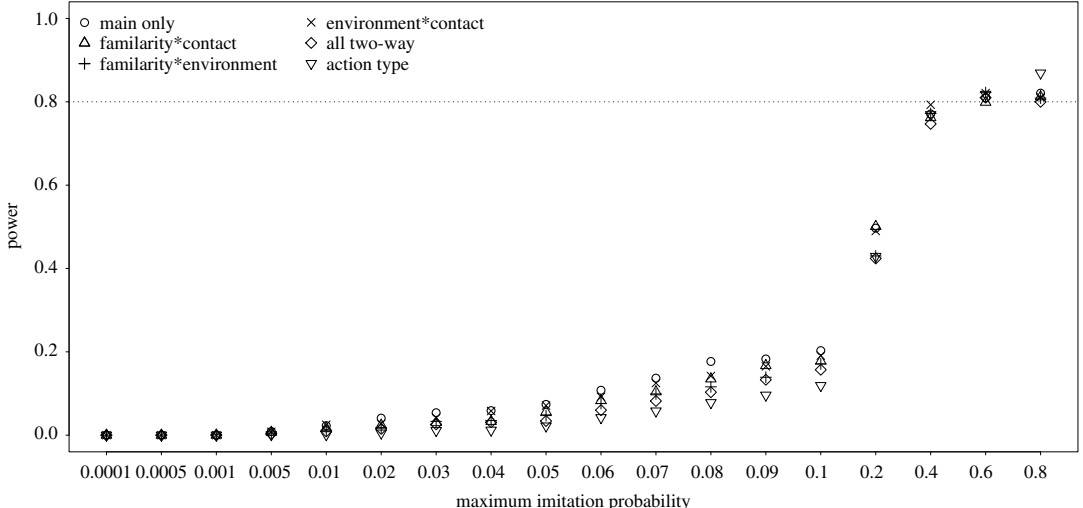

**Figure 1.** Probability of a significant full–null model comparison (power) for simulated data ($N = 1000$ datasets) with different maximum imitation probabilities. The models are labelled according to which two-way interactions they comprised (if any), whereby the model labelled as 'action type' comprised a factor with one unique level for each particular combination of levels of the factors action familiarity, contact involvement and environment.

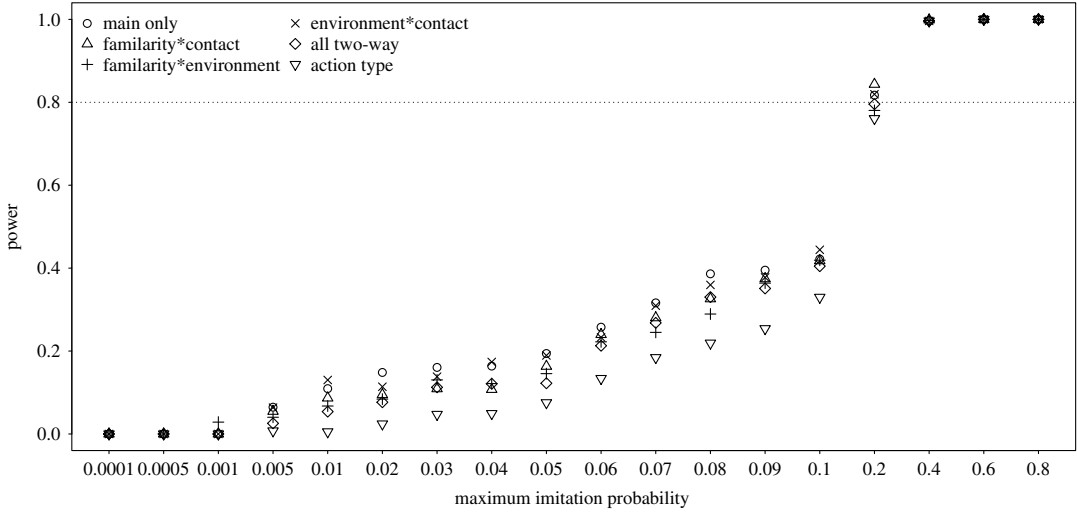

**Figure 2.** Probability of a significant full–null model comparison (power) for simulated data with different maximum imitation probabilities, whereby we considered only datasets for which the full and null model converged without any warning. The models are labelled according to which two-way interactions they comprised (if any), whereby the model labelled as 'action type' comprised a factor with one unique level for each particular combination of levels of the factors familiarity, contact and environment.

differ much between models comprising only main effects and the three models each comprising one two-way interaction.

The power of detecting the two-way interactions was generally low and barely exceeded 0.2 for the two-way interaction between familiarity and contact, when the maximum imitation probability was 0.4 or more (figure 5). When there was a detectable difference in power between a model comprising only one two-way interaction and the model comprising all three two-way interactions, it was slightly higher for the latter. The model with action type as the sole predictor parametrizing the different combinations of actions had by far the highest power which also exceeded 0.8 when the maximum imitation probability was at least 0.4.

Finally, we evaluated which of the six models was the best according to Akaike's information criterion, considering only simulated datasets for which all six full models converged. We found that across all simulated maximum imitation probabilities, the model including only the main effects of

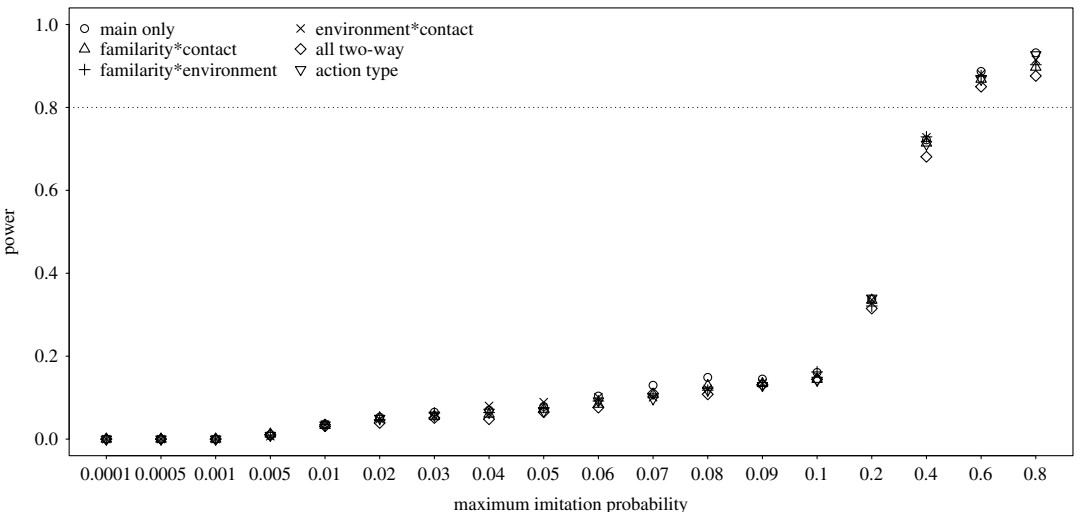

**Figure 3.** Probability of the effect of rearing to reveal significance (power) for simulated data with different maximum imitation probabilities, whereby we considered only datasets for which the full model converged without any warning. The models are labelled according to which two-way interactions they comprised (if any), whereby the model labelled as 'action type' comprised a factor with one unique level for each particular combination of levels of the factors familiarity, contact and environment.

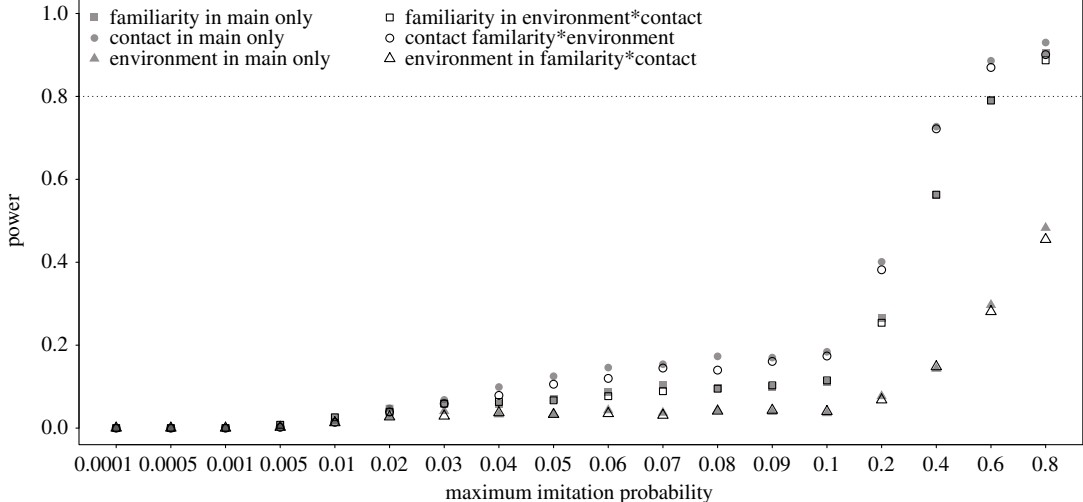

**Figure 4.** Probability of the main effects of familiarity, contact and environment to reveal significance (power) for simulated data with different maximum imitation probabilities, whereby we considered only datasets for which the full model converged without any warning. The models are labelled according to which two-way interactions they comprised (if any).

familiarity, environment and contact was most commonly chosen as the best model (figure 6). However, when the maximum imitation probability was 0.2 or higher, a considerable fraction of the best models were those that included the two-way interaction between familiarity and environment or the two-way interaction between familiarity and contact.

As we did not know how the actual imitation probabilities differed between the cells of our design and in which cell we would find the maximum imitation probability, we also determined the number of imitation events for each of the simulated datasets for which at least one of the full models converged. For the maximum imitation probability of 0.4 (for which most of our assessments of power revealed a sufficient power of approx. 0.8), we found that the simulated datasets comprised an average of 67 imitation events (median; range: 46–90; quartiles: 62 and 73). Given the results of this simulation, we could only fit a model if we found at least 46 imitation events in the dataset. Since we hypothesized that the three action-specific factors (familiarity, environmental effect and contact involvement) interact with one another and given that we could not include their three-way interaction in our models, our

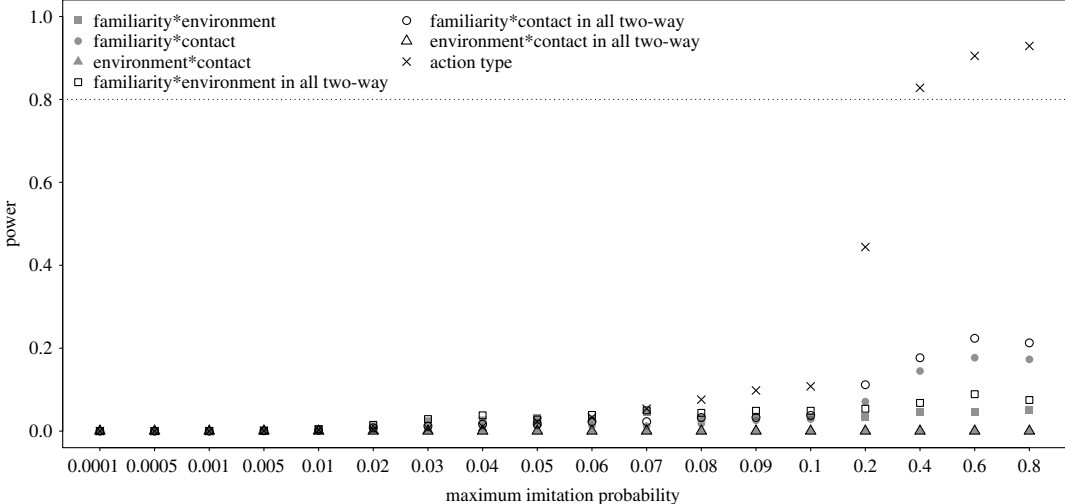

**Figure 5.** Probability of the three two-way interactions between familiarity, contact, environment and action type (the factor comprising one level for each combination of familiarity, environmental effect and contact involvement) to reveal significance (power) for simulated data with different maximum imitation probabilities, whereby we considered only datasets for which the full model converged without any warning. The models are labelled according to which two-way interactions they comprised (if any), and the model labelled as 'action type' comprised a factor with one unique level for each particular combination of levels of the factors familiarity, contact and environment.

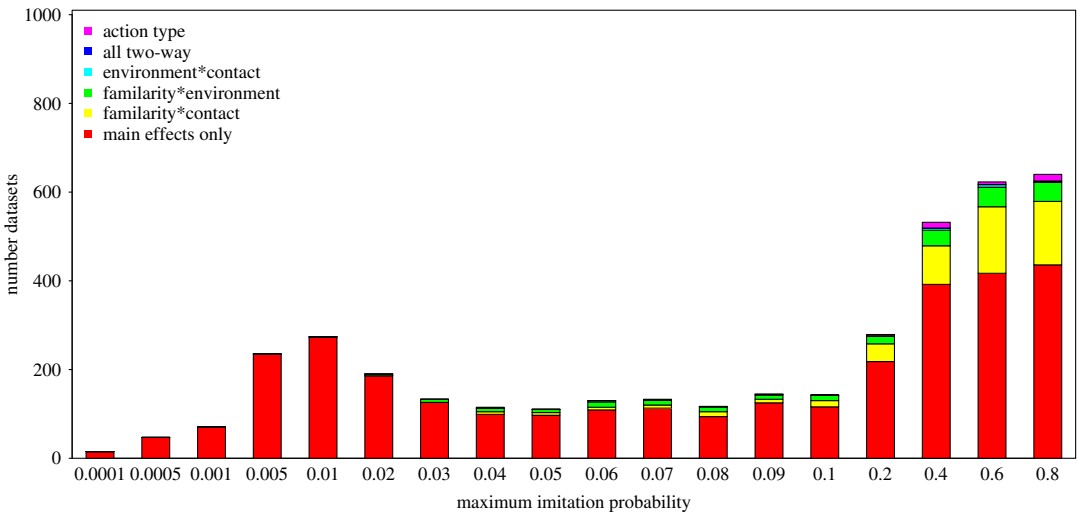

**Figure 6.** Best model according to Akaike's information criterion corrected for small sample size for simulated data with different maximum imitation probabilities when we considered only datasets for which all six full models converged without any warning. The models are labelled according to which two-way interactions they comprised (if any), and the model labelled as 'action type' comprised a factor with one unique level for each particular combination of levels of the factors familiarity, contact and environment present in the data.

initial full model comprised the factor action type (including one level for each combination of familiarity, environmental effect and contact involvement), which was also the parametrization of the interactions among the factors for which we found the highest power (figure 5).[5] In addition to action type, the full model comprised all other predictors present in the models evaluated in this simulation (rearing history, age, age-squared, trial, demonstration number and their interaction) and a random slopes structure identical to the one we included in the action type model. With this single model, we could

---

[5]If the full–null model comparison had been significant and we had found an effect of action type, we would have then evaluated the results of this model and compared its fit with that of simpler models by means of AICs.

address simultaneously the four hypotheses of the Demonstration experiment (effects of action- and subject-specific factors) as well as control for other effects potentially present in the data.

## 2.5. Ethics statement

### 2.5.1. Participation

The study presented here was a non-invasive observational study in which the participation was voluntary. Visitors gave their written consent for their inclusion in the study (see attached 'Declaration of consent' in the electronic supplementary material, S1). Their participation was equivalent to a normal zoo visit. The participation of the chimpanzees in the study was also voluntary. In the Demonstration experiment, only chimpanzees that voluntarily entered the sleeping rooms were tested. No changes in feeding or housing routines were applied due to this study.

### 2.5.2. Management of human participants' data

The methodology described in this manuscript regarding the treatment of human data was approved by the Ethics Commission of the University of Tübingen on 18 June 2018. In the visitor condition, human participants were free to stop their participation when they wished to do so, as no minimum participation time was required. If any participant had wished to withdraw from the experiment, their video clips would have been deleted immediately and their paper questionnaires would have been destroyed. Participants were only included in the study if they agreed to sign both a 'Declaration of consent' and 'Declaration of consent for video and audio recordings' (see documents attached in the electronic supplementary material, S1). All participants were assigned a participation ID which was used to identify each participant's videos and written forms. We ensured the privacy of the participants' data by not linking participants' names with their questionnaires—but only their IDs. Moreover, Name-ID information data were only available to the PI (C.T.) and the main experimenter (A.M.-R.) and were stored securely at the University of Tübingen. Paper data records of Name-ID links were kept locked in the office of the PI at the University of Tübingen. Video recordings were locked safely in the office or—securely—on the university computer network (i.e. not accessible outside the research group). Any personal details were kept separately, and locked or encrypted and password-protected so the participants names were not linkable to either the data or video recordings. Only members and collaborators of the research lab STONECULT were allowed to access the pseudonymized data of the 'Visitors condition'. If videos derived from the study are presented to the public, all participants' faces will be covered before public presentation and data will be, therefore, anonymous. In line with the Data Protection Act 1998 (https://www.legislation.gov.uk/ukpga/1998/29/contents), which makes special provision for personal data collected for research purposes, and the General Data Protection Regulation (https://eur-lex.europa.eu/legal-content/EN/TXT/PDF/?uri=CELEX:32016R0679&from=EN), we will store the data and personal information for 10 years following the publication of empirical articles or communications. After this time, both the videos and the coding list will be deleted. The participants will have the right to ask for the videos where they appear to be deleted at any time before this moment and consequently, the coding list will be kept for as long as the videos exist.

# 3. Results

## 3.1. Visitor experiment

Fifty first-time zoo visitors were tested in the Visitor experiment (28 women, 21 men and one participant that chose not to specify a gender). The median decade when the participants were born was in the 1980s (range of decades 1950–1990). The visitors participated in the experiment for a total time of 5 h 30 min and 58 s (mean duration of interaction per visitor 6 min and 37 s).

Thirty potential imitative events between visitors and chimpanzees were live coded. These potential imitative events took place during the trials of nine different visitors (range of potential imitative events per visitor 1–8). The average duration of the visitors' engagement when potential imitative events took place was 14 min and 4 s (duration range 1 min and 57 s to 45 min and 12 s). The total time spent by the chimpanzees within 2 m of the zoo visitors (the target of our observations) was 23 min and 46 s. We

found substantial inter-rater agreement regarding the presence or absence of action matching in the potential imitative events (Cohen's $\kappa = 0.63$; [25]).

The potential imitative events involved several actions, which were occasionally combined and produced simultaneously. Humans initiated potential imitative events with actions such as knocking or tapping on the window ($N = 5$); loudly vocalizing (e.g. saying hello or reproducing a pant hoot; $N = 6$); making a non-voiced sound (e.g. whistle, kiss sound, click tongue; $N = 4$); putting the palm on the window without making a noise ($N = 3$) and waving ($N = 4$). Chimpanzees initiated potential imitative events performing actions such as raising the chin ($N = 1$); raising the arm ($N = 1$); walking with their flank touching the window ($N = 1$) and kicking the window ($N = 1$). However, it is worth noting that with our experimental set-up, it was not possible to determine if the actions performed by the chimpanzees in potential imitative events were meant to elicit responses in the human visitors (i.e. the chimpanzees could have started walking in short proximity to the visitor for multiple reasons).

From the 30 potential imitative events detected during live coding, the experimenter coded four cases of action matching whereas the second coder confirmed only two of these four instances, rendering the other two observations of action matching ambiguous (table 4). In the two events where action matching was coded by the two independent coders, the reproduction was deemed as full, meaning that both actions and body parts involved were matched. One of the two imitative events was initiated by a chimpanzee (Lutz, male, 15 years old at the time of testing), while the other was initiated by a visitor. The first of these events involved a chimpanzee raising up the chin (which was matched by the visitor) while the latter involved a human pressing his knuckles against the glass window, which was matched by the chimpanzee (Leon, male, 7 years old at the time of testing). The clips of these two events can be found in the OSF link https://osf.io/yar9f/?view_only=17275ce00c0246669bb9353e1669fe21.

Regarding the species that initiated the potential imitative events, we found substantial inter-rater agreement between the two coders (Cohen's $\kappa = 0.84$, [25]). In 26 cases, both coders agreed on the species that initiated the potential imitative event, in one case, the coders disagreed and in three cases, the second coder was unsure or the actions were out of camera frame (coded as NA). Within those cases were both coders agreed ($N = 26$), we found that in 88% of the events, it was the human visitors who initiated the imitative events. In the 12% of cases where both coders agree that chimpanzees 'initiated' the imitative events ($N = 3$), all chimpanzees were mother-reared.

Regarding the perception of the visitors during the potential imitative events, none of the nine visitors that participated in these events perceived that the chimpanzees imitated them. Three out of the nine visitors involved in potential imitative events reported that they performed actions that they had previously seen in the chimpanzees. Interestingly, only during the trial of one of these three visitors, the coders coded action matching initiated by the chimpanzee. Therefore, it is possible that the other two visitors did reproduce chimpanzee actions but outside potential imitative events (perhaps as attention seekers).

## 3.2. Demonstration experiment

Out of the 336 trials conducted during the demonstration experiment, two had to be excluded given that the chimpanzee moved out of camera frame and his/her responses could not be filmed. Inter-rater agreement between the two coders on the presence of action matching was found to be fair (Cohen's $\kappa = 0.38$; [25]). Both coders agreed regarding the presence of action matching in five trials. The experimenter coded a total of 15 cases of action matching (4% of trials, all in a short time frame), whereas the second coder coded a total of 10 cases of action matching (i.e. 3% of trials; table 4).

Of the 15 trials for which action matching was coded by the experimenter, 14 were considered by the experimenter to be partial action matching. Partial action matching was coded when the chimpanzee followed a demonstration by performing an action that did not match the demonstration but that involved the same body part as the demonstrated action ($N = 13$) or when the chimpanzee matched the demonstrated action but using another body part ($N = 1$). In this last case, the chimpanzee was demonstrated the 'touch own shoulder action' and she placed the hand under her armpit (thus the target of the behaviour was not the demonstrated one but the physical actions matched those demonstrated). Ten of the 14 partial action matches (71%) involved familiar actions, nine (64%) produced environmental effects and all (100%) involved contact. From the trials where both coders identified action matching ($N = 5$), both coders agreed in four of these five trials that the accuracy of the match was only partial.

Full action matching was coded by the experimenter in one out of the 334 trials and this instance involved tapping. The chimpanzee involved in this trial was human-reared and had started tapping

**Table 4.** Summary of results from the two experiments conducted. 'N' refers to the total number of chimpanzees tested in each experiment; 'total' refers to the total number of visitors tested or trials conducted. The subsequent columns illustrate the number of total, partial and full action copying events coded by the experimenter (first coder) and by the second coder as well as the number of events coded equally by the two coders (columns 'first and second coder agree').

| experiment | N | total | action copying | | | partial action copying | | | full action copying | | |
|---|---|---|---|---|---|---|---|---|---|---|---|
| | | | first coder | second coder | first and second coder agree | first coder | second coder | first and second coder agree | first coder | second coder | first and second coder agree |
| Visitor experiment | 32 | 50 visitors | 4 | 2 | 2 | 1 | — | | 3 | — | |
| Demonstration experiment | 14 | 334 trials | 15 | 10 | 5 | 14 | 9 | 4 | 1 | 1 | 0 |

as a begging gesture before the demonstration took place (experimenter personal observation). The second coder also reported a single case of full action matching in a trial, but this was a different case from the one coded by the experimenter (the experimenter coded this instance as partial action matching, table 4). The human-reared chimpanzee involved in this trial briefly touched her shoulder after being demonstrated the 'touch own shoulder' action.

We did not find enough action matching events to be able to fit a model with enough statistical power to evaluate which factors influence the probability that chimpanzees imitate demonstrated actions (our estimated threshold was 46 trials with partial or full action matching). Our estimated threshold was calculated using a simulation with 16 individuals rather than the 14 that we tested (after one hand-reared chimpanzee died shortly before the experiments began and to maintain the number of chimpanzees in both rearing groups equal, one mother-reared individual was also excluded). However, we do not expect that a recalculation of the threshold adjusting the number of individuals would lower the threshold enough to be able to run our proposed model: the number of action matching events detected would be between five (if we exclude cases were both coders disagreed) and 15 (if we take the cases of action matching coded by the experimenter), both numbers considerably below our previously calculated threshold. Consequently, we did not repeat our simulation.

# 4. Discussion

In this study, we investigated interspecies imitation between chimpanzees and humans using two complementary approaches. On the one hand, during the Visitor experiment, we assessed the frequency of imitation events and action matching between first-time zoo visitors and group-housed chimpanzees. On the other hand, during the Demonstration experiment, we investigated chimpanzee's imitative abilities when tested individually and after being provided with demonstrations of actions with different characteristics (e.g. familiarity, anchoring, environmental effect and contact). Overall, we did not find convincing evidence of chimpanzees' imitative abilities of human-demonstrated actions beyond some anecdotal observations.

Our first aim in the Visitor experiment was to assess whether chimpanzees and/or zoo visitors spontaneously imitate any actions of the other species. We found that potential imitative events (when the chimpanzees and visitors were less than 2 m apart through the glass window) were very rare. Even though we instructed the zoo visitors to try to make the chimpanzees imitate them (and consequently the visitors gestured and performed a wide variety of actions), the chimpanzees hardly ever came in close proximity with the visitors. In the few occasions when they did ($N = 30$), the chimpanzees rarely performed any actions—let alone matching actions—in response to the actions performed by the visitors. The chimpanzees very rarely performed any actions directed to the zoo visitors ($N = 3$). In addition, only in two potential imitative events, the two coders agreed that action matching took place (one initiated by a chimpanzee and one initiated by a visitor). The second aim of the Visitor experiment was to test whether human biases exist in perceiving that the chimpanzees imitate visitors even when action matching does not actually take place. On this regard, we found that there was strong inter-observer agreement between the experimenter who live-coded the imitative events and the second coder, who only disagreed in two potential imitative events. Therefore, our results suggest that the live-coder was not biased towards perceiving imitation.

Taken together, our results show that spontaneous interspecies imitation between chimpanzees and zoo visitors is extremely rare. Our findings are in stark contrast with a previous study investigating spontaneous interspecies imitation between five captive chimpanzees and zoo visitors [16]. Persson et al. [16] found frequent interspecies imitation between both species at similar rates, with zoo visitors engaging in more accurate action matching than the chimpanzees. Although our total observation time was smaller than that reported by Persson et al. [16] (partly due to our stricter data collection method), we found much lower rates of action matching (and even interspecies interaction) than those reported by Persson et al. [16]. This disparity in results could be the consequence of a variety of factors (see Introduction section) including the rearing background of the chimpanzees, the design of the enclosure, the chimpanzee group composition or the fact that we tested zoo visitors individually. Each of these factors could have influenced how often the chimpanzees interacted with the zoo visitors. Persson et al. [16] only tested five chimpanzees, four of which had been at least partly human-reared. In our study, we tested 32 chimpanzees, only seven of which had been partly human-reared. Human-reared individuals have been shown to be more human-oriented than mother-reared

individuals [2] as well as to present higher imitative abilities [10,26]. Given that almost the entire group of chimpanzees tested by Persson *et al.* had been human-reared, it is possible that these individuals were more oriented towards the visitors than the chimpanzees included in our sample as well as presented higher imitative abilities due to their rearing background.

Moreover, the present experiment was conducted in the large outdoor enclosure of the chimpanzees, whereas Persson *et al.*'s study was conducted both indoors and outdoors, with close interspecies interactions (less than 2 m) taking place in the indoor enclosure. Given that the outdoor enclosure of the chimpanzees at Leintal zoo is probably larger that the indoor enclosure of the chimpanzees tested by Persson *et al.*, it is possible that the enclosure size affected how often the chimpanzees were in close proximity to the zoo visitors (although note that [16] did not report enclosure sizes and, therefore, this is just a hypothesis). Another factor influencing interaction rates between the chimpanzees and the visitors might have been the group size. The larger group tested in our study compared with Persson *et al.*'s study might have caused a reduction in the attention that the chimpanzees paid to the zoo visitors in our experiment (perhaps compensating for a potential higher tendency to imitate of the hand-reared individuals included in our study, see also Demonstration experiment). The fact that we tested individual zoo visitors rather than groups as in the study by Persson *et al.* [16] could have influenced the time the visitors participated in the experiment and consequently the interspecies interaction rates. However, testing individual visitors was a necessary measure in order to prevent pseudoreplication in the data, as testing several participants at once might have led to the participants influencing each other's behaviour.

Furthermore, Persson *et al.* live-coded interactions between the chimpanzees and groups of visitors, whereas in our study, we tested single zoo visitors. Even coding potential imitative events between a single zoo visitor and the group of chimpanzees proved challenging. Interactions often occurred rapidly, involving several chimpanzees at once, and the visitors often performed several actions simultaneously (e.g. waving the arms while saying hello). Live-coding such events with groups of visitors rather than a single individual must have been very challenging and might have led to an overestimation of action matching rates (however, note that this hypothesis cannot be tested as no video recordings were produced by Persson *et al.* [16]).

The aims of the Demonstration experiment were twofold. First, we wanted to investigate if chimpanzees would imitate actions performed by human demonstrators when tested individually. Second, if they did so, we wanted to model which action- and subject-specific factors elicited imitation. Using a simulation, we estimated that in order to address our second aim with enough statistical power we needed to identify at least 46 imitation events. Although this number would need to be slightly adjusted due to the death of a chimpanzee included in our original estimation, we did not find enough imitative events to apply our model. Furthermore, given that the actual number of imitations we found was very low and limits model complexity to an extent that even the simplest of our models could not possibly be fitted, we refrained from repeating the power analysis with the smaller number of chimpanzees we tested. Both coders agreed on the presence of action matching in only five action demonstrations and the two coders reported 15 and 10 action matching events, respectively, out of the 334 action demonstrations conducted. Furthermore, the vast majority of action matching reported was partial—i.e. the specific action demonstrated was not matched in full (except in two cases; see Results). Instead, the vast majority of events involved partial action matching, i.e. the chimpanzees performed an action different from the demonstrated action that involved the same body part included in the demonstration (after [10]). Therefore, copying of actions was absent in the tested chimpanzee population (except for two out of 334 cases which are ambiguous at best, given the lack of inter-observer agreement).

Regarding the few cases of action matching that we detected across both experiments, it is possible that these observations represent examples of action copying. However, alternative explanations should also be considered as action matching could take place as a consequence of other phenomena different from copying. First, the action matching events observed in our experiments could be due to chance and not elicited by the observation of human demonstrations. Second, response facilitation has been shown to elicit familiar actions present in the subjects' repertoire after the observation of another individual performing the behaviour (e.g. yawning; [27,28]). Third, local enhancement could have led the chimpanzees to touch the same location that the demonstrator was taping (Demonstration experiment) or putting her hand on the window (Visitor experiment) without the chimpanzees necessarily copying the demonstrated action. Future studies could exclude these possibilities by first calculating base rates at which different actions take place and then comparing these base rates with the rates at which these actions are performed after observing demonstrations of these same actions as

well as demonstrations of other different actions (for a detailed explanation of this 'cross-target' approach and its application to the neonatal imitation literature, see [29]).

Overall, the combined results of the two experiments presented here show that chimpanzees' action matching, in which chimpanzees reproduce actions performed by human demonstrators, is extremely rare. In over 20 h of human–chimpanzee interactions, we found a single instance (detected by two independent coders) of full action matching by a chimpanzee. Similarly, after 334 demonstrations of actions with different characteristics performed by a human, we found a single instance of full action matching (not confirmed by the second coder). This absence of action matching was independent of the chimpanzee's rearing background, the familiarity of the demonstrated actions, the presence or absence of an environmental effect and the involvement of physical contact (i.e. anchoring) in the demonstrated actions. Our results are in line with previous studies reporting the absence of action copying in chimpanzees (e.g. [2,14,30]) and other great ape species (bonobos: [15]). However, our results are in contrast with previous studies reporting action copying in enculturated/human-reared chimpanzees [2,26,31], an enculturated orangutan [10] and chimpanzees trained in particular tasks (Do-as-I-Do; [7,9]). Such differences between studies might be due to varying degrees of enculturation, specific training (and consequently human orientation; [7]) or housing conditions among the tested animals. Although future studies are necessary to discern the factors that led to action copying in the individuals included in these specific experiments [2,10,26], our results question that action matching is widespread among group-housed chimpanzees.

Data accessibility. The code employed to conduct the simulation and data analysis as well as all our data and approved Stage 1 protocol (including data predictions; https://doi.org/10.17605/OSF.IO/C5J28) can be found in the OSF project link https://osf.io/yar9f/?view_only=17275ce00c0246669bb9353e1669fe21. We confirm that data collection and analyses were conducted following the approved protocol.

Authors' contributions. A.M.-R. and C.T. designed the study. A.M.-R. collected the data. A.M.-R. and R.M. analysed the data. A.M.-R., C.T., R.M. and J.C. wrote the manuscript.

Competing interests. We declare we have no competing interests.

Funding. This work was supported by the Excellence Initiative of the University of Tübingen.

Acknowledgements. The authors are grateful to Leintal zoo for allowing and helping us to conduct the observations and experiments described in this manuscript. The authors thank David Boysen for his help with data collection and Maribel Rodrigo Aleixandre for acting as second coder. We are also thankful to Dr Lydia Hopper, Damien Neadle, William Snyder and Jordy D. Orellana for comments on previous versions of this manuscript.

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
