## [Peer Review File · Royal Society Open Science]

Review History

RSOS-200228.R0 (Original submission)

Review form: Reviewer 1

Do you have any ethical concerns with this paper?

No

Recommendation?

Accept with minor revision

Comments to the Author(s)

Please find my comments regarding each of the key aspects of the proposal below.

The scientific validity of the research question(s)

The authors outline a planned study of imitation in captive chimpanzees, using two experimental approaches. The first will attempt to replicate (with refinements to the methodology) results from Persson et al., who report imitation between chimpanzees and zoo visitors. The second experiment will investigate the potential factors which contribute to imitative abilities in

chimpanzees (such as enculturation, familiarity of the action), using individual testing of captive chimpanzees.

The questions this study aims to address are clearly explained and are scientifically valid. Replicating the Persson et al. study is a good idea (with the proposed addition of video recording to strengthen the methodology). The second experiment would be a useful addition to the literature by considering together multiple factors impacting imitation.

The logic, rationale, and plausibility of the proposed hypotheses

It might be useful if the authors could add explicit hypotheses alongside the aims of their studies. Their data predictions appear to me to be well reasoned and clear.

The soundness and feasibility of the methodology and analysis pipeline (including statistical power analysis where applicable)

Visitor experiment:

You mention a planned sample size of 25 participants for ~15 hours of interactions. This implies around 30 minutes of interaction time per participant, and given you won't impose a minimum interaction time, I think that may be a little long to expect people to stay for?

I would consider clarifying in your planned methods whether your goal in terms of the sample size for this study is the number of participants, or the number of hours of interaction time - my suspicion is you may need to recruit more than 25 participants to reach 15 hours of interactions. I would also suggest planning to recruit a larger sample of participants as you may find substantial inter-individual variation in the behaviours the human participants attempt, and this would also give you a larger sample for your interesting question regarding humans' potential bias towards perceiving imitation.

Demonstration experiment:

The choice of actions seems sensible to me, and the choice of both contact / non-contact actions, and actions with and without environmental effects is good.

How will you select the 8 mother-reared individuals? Will they be age and sex matched (as far as possible) with the hand-reared individuals? Or the individuals considered most likely to engage in the testing process? Or randomly selected?

The coding schemes for both experiments seem well-planned and thorough. I think there's a typo in the Demonstration coding scheme, it reads "the chimpanzees or the humans perform an action after an action has been demonstrated." In the Demonstration experiment, only the chimpanzees could imitate, correct? The human demonstrators presumably will be instructed not to imitate the chimpanzees in this experiment?

You may wish to also code for the identity of the demonstrator in these trials, if there's the possibility of using different demonstrators. Some chimpanzees may have closer relationships with some keepers than others, and I can imagine this could impact their level of attention to the demonstrations at least.

The power analysis / simulation is very convincing. It isn't clear to me from the current report what your approach will be if you find more than 0, but fewer than 46 imitation events in your data set - you state you will only fit your full model with 46 events, in order to have good power. Would you attempt any alternative statistical analysis with fewer than 46 events? Would you be able to draw any conclusions from fewer than 46 events - from your power analysis I understand it would not be possible to explore the factors contributing to imitation - how will you then present the results?

Whether the clarity and degree of methodological detail would be sufficient to replicate exactly the proposed experimental procedures and analysis pipeline

If the above comments are addressed, I believe the methods and analysis could be replicated by others.

Whether the authors provide a sufficiently clear and detailed description of the methods to prevent undisclosed flexibility in the experimental procedures or analysis pipeline
I think the Visitor experiment needs a little clarification regarding the 'stopping point', whether this will be the number of participants or hours of interaction time, but aside from this I think the methodology for this study is clear.

Whether the authors have considered sufficient outcome-neutral conditions (e.g. positive controls) for ensuring that the results obtained are able to test the stated hypotheses
The study seems well planned. The planned use of double-coding from video, with paired dummy examples, seems a sufficient quality control.

Additional comments -

Participant info sheet / consent forms

- Typo in final paragraph of info sheet (switches from 'you will' to 'my name will never appear')
- This may be a non-issue following translation into German, but I would suggest giving a very brief explanation of what imitation is using lay terms (e.g. "we're interested in seeing whether the apes will copy you"). I would consider 'imitate' a somewhat technical term in English and would simplify it for zoo visitors – but perhaps this won't be a problem once the document is translated.
- I would consider adding something to the info sheet to discourage the potential bad behaviours you list in your methods – I think it's good that you plan to end the trial if a participant bangs loudly on the glass, for example, but it might be even better to explicitly discourage this behaviour before you start testing. I don't think this would place too great a limitation on the behaviours participants perform when trying to interacting with the apes.

Review form: Reviewer 2

Do you have any ethical concerns with this paper?

No

Recommendation?

Accept with minor revision

Comments to the Author(s)

Overall, I found this to be a clear and robust proposal for an important piece of work. The authors propose a replication of a contentious study by Persson et al. (2017) studying between-species imitation of chimpanzees and humans, as well as exploring interesting additional questions regarding human bias towards perception of imitative events. Crucially, the authors propose considerable methodological refinements to the methods used by Persson et al. through the use of video recordings and multiple coders as well as creating a 'behavioural baseline' for their chimpanzee sample, to ensure that any seemingly imitative behaviours are indeed novel.

A particular strength of the proposal is that the authors' plan to take a 'two prong' approach, using side-by-side observational and experimental designs. The second, experimental portion of the study will test for human-directed imitation in chimpanzees under a more controlled setting. The authors are interested in exploring the various individual and action-related factors that

influence the likelihood with which imitation events occur. The authors have approached this portion of the study with a level of rigour and preparation which is rare in the field.

I have three main issues which the authors should address:

- 1) While the authors carried out careful power calculations for the demonstration experiment, I have major concerns regarding the proposed sampling size and logistics for the observational study.
- 2) The use of 'finger pop' as the candidate for imitative behaviour in the demonstration experiment is problematic.
- 3) There are two potentially significant but easily resolved issues regarding the behavioural ethogram proposed for determining a behavioural baseline for the chimpanzees. Firstly, a denial of the significance of individual differences in behavioural repertoires for a study of this kind and secondly a lack of clear definition on many of the items in their ethogram.

I have detailed each of these major comments below, as well as a handful of more minor comments.

I would emphasise that each of the issues I have raised would simply require refinement of the proposed methods rather than any sort of overhaul, and would not anticipate that they be barriers to completion of the work or its subsequent publication.

Major comments:

Sampling

In my experience of zoos, it seems unlikely that visitors will spend, on average, 30+ minutes observing the chimpanzees. The public tends to stick around for long periods during events such as feeding or when the animals are otherwise particularly active (and therefore perhaps less likely to interact with the public), but during rest periods may only spend a few minutes waiting for something interesting to happen and then leave. The researchers should therefore consider how they will choose the timing of their sampling periods, and what effects this might have in terms of the likelihood that individuals will interact with the public (e.g. during scatter feeds, I would expect zero interaction events).

In light of the above the researchers may need to revise either a) their expectations for the number of hours they will sample, or b) their target number of participants. If not, they should decide whether they will stop at 25 participants regardless of how many hours of interaction data they have collected (if the researchers are 'unlucky' they could easily sample 25 individuals who each spent less than 5 minutes in the exhibit or are never afforded an opportunity to 'interact' with a chimpanzee), or continue sampling until a secondary criteria of time is met.

A more specific description of how sampling will take place is also important. Will multiple individuals from a group of visitors be sampled? If so, can these be considered independent data points? If not, are we to assume that participants will be taken to a more private area, since non-consenting individuals cannot be recorded on video? The authors may wish to consider how this will influence a) how long participants will spend at the enclosure (probably not long, if their family/friends are waiting for them) and b) how this might influence chimpanzee behaviour (are they more likely to spend time at windows with more people present?).

While I think the use of video recordings will yield high quality data in the proposed study, it is also a limiting factor. Persson et al.'s methods were no doubt noisier, but had the advantage of sampling ALL individuals who passed through the exhibit during an extensive sampling period. To illustrate: 52 hours of observation in Persson et al. (2017) in which they were able to sample interactions between the chimps and let's say, conservatively, 10 individuals visiting at any given moment (it was during the busiest periods) - effectively yielding 520 hours of potential interaction time. They recorded 3794 interaction observations, 10% of which were reported to be imitative: a rate of 0.7 potentially imitative acts per person per hour. Assuming this figure to be accurate, and that approximate interaction rates are consistent between study sites, how many observations of interest would we expect from a replication of just 15 hours from 25 individuals? I am not convinced by the utility of such a small sample.

Methods: "Finger pop sound (hook finger inside mouth and release with a sound)" as action demonstrated in second experiment.

Are there any recorded examples of a chimpanzee doing this? I'm not certain that it is physiologically possible for them to produce such a sound as it requires quite a bit of control over the articulators (lips + cheek) and airflow in order to achieve an audible 'pop', which are motor functions not generally thought to be under much top-down control by chimpanzees.

The opacity of the causal factors in achieving this sound also make it a poor candidate for examining imitation. Indeed, the authors include a link to a 'wiki how' article providing instructions on how to perform a finger-pop and a quick google search brings up a list of similar articles and Youtube videos (in a quick survey of my friends I found 2 in 10 could not, and another could only do so with practice). If this degree of explicit verbal instruction is necessary for some humans, it does not seem a 'fair' behaviour for use here. Unless the authors have a strong justification for this choice, therefore, I strongly recommend choosing another behaviour.

The ethogram

Firstly: The authors state that collecting individual action repertoires is not necessary or reasonable (Footnote 1). I am surprised by this - anyone who has worked with captive chimps is likely to be familiar with individuals who demonstrate idiosyncratic behaviours (usually when interacting with humans). Individual variation in behavioural repertoires is therefore quite crucial to establishing a really watertight behavioural baseline for a study like this, and should be acknowledged as a limitation of the design if it is not possible. The authors appear to have identified a couple of candidates for this in their ethogram ("window cleaning" and "raspberry") - it may be worth at least asking carestaff if they have observed these in all or most individuals. They could also be asked if there are any other unusual behaviours which are currently missing from the ethogram.

Secondly: The entries in the ethogram taken from Persson et al. (2017) are not useful. Currently, the authors have either interpreted their meaning independently ("There are no definitions of this behaviors in original paper, interpreted as chimpanzee makes short audible contact between knuckles and head"), or not defined them at all, both of which are inadequate. Why not contact the original authors and request proper definitions? Such a vague ethogram is not acceptable when the exact form of behaviours is so crucial to the research question at hand.

Minor comments:

The questionnaire: I think the correlation of video data with questionnaire reports on imitative behaviour from the public will be interesting. However, I would caution against the implication that any bias identified here is likely to explain the results of Persson et al. (2017) - the

Line 281: “eight hand-reared individuals (i.e., enculturated to some degree)”

More details on what hand-rearing entailed in this context should be gathered if the authors wish to include this variable in their analysis. i.e. Were they raised in human homes, a lab, or a nursery? Were they raised alongside other chimpanzees? How many hours of human contact did they receive per day? How long were they hand-reared for? Lumping all hand-reared individuals (which could mean almost anything from ‘grew up in a human household and was dressed up like a sailor’ to ‘was bottle-fed by a keeper twice per day in a chimpanzee nursery’) in the same category as truly enculturated apes (Kanzi, Ai, Lucy, etc.) is unlikely to be informative.

Line 310: “A potential imitative event will be considered as each case when a chimpanzee and a visitor interact through a glass window or the mesh of the outdoor enclosure.”

How will an ‘interaction’ be defined?

Line 323: “If the demonstrated action has an environmental effect (e.g. creates a sound) or not.”

Just as a consideration: Zoo enclosure glass is generally very thick and may block out anything less noisy than a chimpanzee display or pant hoot.

Line 328: “From the video recordings compiled during the visitor experiment, the presence of action matching and the initiating species will be coded from each potentially imitative event a second time in order to determine if the experimenter was biased regarding the perception of imitative events, particularly the species that initiates such events.”

Will the experimenter and the individual doing the second coding be the same person? What will happen if the two coders disagree with each other?

Will interactions not coded as ‘potentially imitative’ by the first coder also be second-coded in this fashion? If not, why not? Giving positive data points two possible opportunities for rejection and negative data points only one is likely to quickly bias the dataset.

Line 440: “We do not expect imitation probability to be affected by age, trial number, or demonstration number, nor do we expect it to vary considerably between individuals”

On the contrary, I would predict large amounts of individual variation according to how human-oriented individuals are. The authors also note elsewhere that they expect that rearing history may cause individual differences in imitative behaviour.

Lines 482-508: These model descriptions are difficult to follow. A table of all model structures used would be much easier to parse.

Lines 707-713: This is good - But what precautions are being taken to avoid false negatives?

Supplemental Material: Reproducibility is contingent on being able to understand what was being done at each step of an analysis, and why. I did not find that I could do either with this script, and feel confident most naive readers would similarly struggle. I recommend it be tidied and appropriately commented.

Decision letter (RSOS-200228.R0)

04-Mar-2020

Dear Ms Motes Rodrigo,

The Editors assigned to your stage one Registered Report ("Evaluating the influence of action- and subject-specific factors on chimpanzee action copying") have now received comments from reviewers. We would like you to revise your paper in accordance with the referee and editors suggestions which can be found below (not including confidential reports to the Editor). Please note this decision does not guarantee eventual acceptance.

Please submit a copy of your revised paper within three weeks (i.e. by the 26-Mar-2020). If we do not hear from you within this time then it will be assumed that the paper has been withdrawn. In exceptional circumstances, extensions may be possible if agreed with the Editorial Office in advance. We do not allow multiple rounds of revision so we urge you to make every effort to fully address all of the comments at this stage. If deemed necessary by the Editors, your manuscript will be sent back to one or more of the original reviewers for assessment.

When submitting your revised manuscript, you must respond to the comments made by the referees and upload a file "Response to Referees" in "Section 2 - File Upload". Please use this to document how you have responded to the comments, and the adjustments you have made. In order to expedite the processing of the revised manuscript, please be as specific as possible in your response.

Kind regards,
Professor Chris Chambers
Royal Society Open Science
openscience@royalsociety.org

on behalf of Professor Chris Chambers (Registered Reports Editor, Royal Society Open Science)
openscience@royalsociety.org

Associate Editor Comments to Author (Professor Chris Chambers):

Comments to the Author:

Two expert reviewers have now assessed the manuscript. Both find significant merit in the proposal while also pointing to a range of issues that will be need to addressed to achieve in principle acceptance, from questions of feasibility (in particular, expectations of visitor observation time - a concern highlighted by both reviewers), methodological detail, sample planning, and presentational clarity. All issues are within the range of concerns that are generally addressable for a Stage 1 RR therefore a Major Revision is invited. Please respond carefully and comprehensively to each point in the reviews.

Comments to Author:

Reviewer: 1

Comments to the Author(s)

Please find my comments regarding each of the key aspects of the proposal below.

· The scientific validity of the research question(s)

The authors outline a planned study of imitation in captive chimpanzees, using two experimental approaches. The first will attempt to replicate (with refinements to the methodology) results from Persson et al., who report imitation between chimpanzees and zoo visitors. The second experiment will investigate the potential factors which contribute to imitative abilities in chimpanzees (such as enculturation, familiarity of the action), using individual testing of captive chimpanzees.

The questions this study aims to address are clearly explained and are scientifically valid.

Replicating the Persson et al. study is a good idea (with the proposed addition of video recording to strengthen the methodology). The second experiment would be a useful addition to the literature by considering together multiple factors impacting imitation.

· The logic, rationale, and plausibility of the proposed hypotheses

It might be useful if the authors could add explicit hypotheses alongside the aims of their studies. Their data predictions appear to me to be well reasoned and clear.

· The soundness and feasibility of the methodology and analysis pipeline (including statistical power analysis where applicable)

Visitor experiment:

You mention a planned sample size of 25 participants for ~15 hours of interactions. This implies around 30 minutes of interaction time per participant, and given you won't impose a minimum interaction time, I think that may be a little long to expect people to stay for?

I would consider clarifying in your planned methods whether your goal in terms of the sample size for this study is the number of participants, or the number of hours of interaction time - my suspicion is you may need to recruit more than 25 participants to reach 15 hours of interactions. I would also suggest planning to recruit a larger sample of participants as you may find substantial inter-individual variation in the behaviours the human participants attempt, and this would also give you a larger sample for your interesting question regarding humans' potential bias towards perceiving imitation.

Demonstration experiment:

The choice of actions seems sensible to me, and the choice of both contact / non-contact actions, and actions with and without environmental effects is good.

How will you select the 8 mother-reared individuals? Will they be age and sex matched (as far as possible) with the hand-reared individuals? Or the individuals considered most likely to engage in the testing process? Or randomly selected?

The coding schemes for both experiments seem well-planned and thorough. I think there's a typo in the Demonstration coding scheme, it reads "the chimpanzees or the humans perform an action after an action has been demonstrated." In the Demonstration experiment, only the chimpanzees could imitate, correct? The human demonstrators presumably will be instructed not to imitate the chimpanzees in this experiment?

You may wish to also code for the identity of the demonstrator is in these trials, if there's the possibility of using different demonstrators. Some chimpanzees may have closer relationships

with some keepers than others, and I can imagine this could impact their level of attention to the demonstrations at least.

The power analysis / simulation is very convincing. It isn't clear to me from the current report what your approach will be if you find more than 0, but fewer than 46 imitation events in your data set – you state you will only fit your full model with 46 events, in order to have good power. Would you attempt any alternative statistical analysis with fewer than 46 events? Would you be able to draw any conclusions from fewer than 46 events – from your power analysis I understand it would not be possible to explore the factors contributing to imitation – how will you then present the results?

- Whether the clarity and degree of methodological detail would be sufficient to replicate exactly the proposed experimental procedures and analysis pipeline

If the above comments are addressed, I believe the methods and analysis could be replicated by others.

- Whether the authors provide a sufficiently clear and detailed description of the methods to prevent undisclosed flexibility in the experimental procedures or analysis pipeline

I think the Visitor experiment needs a little clarification regarding the 'stopping point', whether this will be the number of participants or hours of interaction time, but aside from this I think the methodology for this study is clear.

- Whether the authors have considered sufficient outcome-neutral conditions (e.g. positive controls) for ensuring that the results obtained are able to test the stated hypotheses

The study seems well planned. The planned use of double-coding from video, with paired dummy examples, seems a sufficient quality control.

Additional comments -

Participant info sheet / consent forms

- Typo in final paragraph of info sheet (switches from 'you will' to 'my name will never appear')

- This may be a non-issue following translation into German, but I would suggest giving a very brief explanation of what imitation is using lay terms (e.g. "we're interested in seeing whether the apes will copy you".) I would consider 'imitate' a somewhat technical term in English and would simplify it for zoo visitors – but perhaps this won't be a problem once the document is translated.

- I would consider adding something to the info sheet to discourage the potential bad behaviours you list in your methods – I think it's good that you plan to end the trial if a participant bangs loudly on the glass, for example, but it might be even better to explicitly discourage this behaviour before you start testing. I don't think this would place too great a limitation on the behaviours participants perform when trying to interacting with the apes.

Reviewer: 2

Comments to the Author(s)

Overall, I found this to be a clear and robust proposal for an important piece of work. The authors propose a replication of a contentious study by Persson et al. (2017) studying between-species imitation of chimpanzees and humans, as well as exploring interesting additional questions regarding human bias towards perception of imitative events. Crucially, the authors propose considerable methodological refinements to the methods used by Persson et al. through the use of video recordings and multiple coders as well as creating a 'behavioural baseline' for their chimpanzee sample, to ensure that any seemingly imitative behaviours are indeed novel.

A particular strength of the proposal is that the authors' plan to take a 'two prong' approach,

using side-by-side observational and experimental designs. The second, experimental portion of the study will test for human-directed imitation in chimpanzees under a more controlled setting. The authors are interested in exploring the various individual and action-related factors that influence the likelihood with which imitation events occur. The authors have approached this portion of the study with a level of rigour and preparation which is rare in the field.

I have three main issues which the authors should address:

- 1) While the authors carried out careful power calculations for the demonstration experiment, I have major concerns regarding the proposed sampling size and logistics for the observational study.
- 2) The use of 'finger pop' as the candidate for imitative behaviour in the demonstration experiment is problematic.
- 3) There are two potentially significant but easily resolved issues regarding the behavioural ethogram proposed for determining a behavioural baseline for the chimpanzees. Firstly, a denial of the significance of individual differences in behavioural repertoires for a study of this kind and secondly a lack of clear definition on many of the items in their ethogram.

I have detailed each of these major comments below, as well as a handful of more minor comments.

I would emphasise that each of the issues I have raised would simply require refinement of the proposed methods rather than any sort of overhaul, and would not anticipate that they be barriers to completion of the work or its subsequent publication.

Major comments:

Sampling

In my experience of zoos, it seems unlikely that visitors will spend, on average, 30+ minutes observing the chimpanzees. The public tends to stick around for long periods during events such as feeding or when the animals are otherwise particularly active (and therefore perhaps less likely to interact with the public), but during rest periods may only spend a few minutes waiting for something interesting to happen and then leave. The researchers should therefore consider how they will choose the timing of their sampling periods, and what effects this might have in terms of the likelihood that individuals will interact with the public (e.g. during scatter feeds, I would expect zero interaction events).

In light of the above the researchers may need to revise either a) their expectations for the number of hours they will sample, or b) their target number of participants. If not, they should decide whether they will stop at 25 participants regardless of how many hours of interaction data they have collected (if the researchers are 'unlucky' they could easily sample 25 individuals who each spent less than 5 minutes in the exhibit or are never afforded an opportunity to 'interact' with a chimpanzee), or continue sampling until a secondary criteria of time is met.

A more specific description of how sampling will take place is also important. Will multiple individuals from a group of visitors be sampled? If so, can these be considered independent data points? If not, are we to assume that participants will be taken to a more private area, since non-consenting individuals cannot be recorded on video? The authors may wish to consider how this will influence a) how long participants will spend at the enclosure (probably not long, if their family/friends are waiting for them) and b) how this might influence chimpanzee behaviour (are they more likely to spend time at windows with more people present?).

While I think the use of video recordings will yield high quality data in the proposed study, it is also a limiting factor. Persson et al.'s methods were no doubt noisier, but had the advantage of sampling ALL individuals who passed through the exhibit during an extensive sampling period. To illustrate: 52 hours of observation in Persson et al. (2017) in which they were able to sample interactions between the chimps and let's say, conservatively, 10 individuals visiting at any given moment (it was during the busiest periods) - effectively yielding 520 hours of potential interaction time. They recorded 3794 interaction observations, 10% of which were reported to be imitative: a rate of 0.7 potentially imitative acts per person per hour. Assuming this figure to be accurate, and that approximate interaction rates are consistent between study sites, how many observations of interest would we expect from a replication of just 15 hours from 25 individuals? I am not convinced by the utility of such a small sample.

Methods: "Finger pop sound (hook finger inside mouth and release with a sound)" as action demonstrated in second experiment.

Are there any recorded examples of a chimpanzee doing this? I'm not certain that it is physiologically possible for them to produce such a sound as it requires quite a bit of control over the articulators (lips + cheek) and airflow in order to achieve an audible 'pop', which are motor functions not generally thought to be under much top-down control by chimpanzees.

The opacity of the causal factors in achieving this sound also make it a poor candidate for examining imitation. Indeed, the authors include a link to a 'wiki how' article providing instructions on how to perform a finger-pop and a quick google search brings up a list of similar articles and Youtube videos (in a quick survey of my friends I found 2 in 10 could not, and another could only do so with practice). If this degree of explicit verbal instruction is necessary for some humans, it does not seem a 'fair' behaviour for use here. Unless the authors have a strong justification for this choice, therefore, I strongly recommend choosing another behaviour.

The ethogram

Firstly: The authors state that collecting individual action repertoires is not necessary or reasonable (Footnote 1). I am surprised by this - anyone who has worked with captive chimps is likely to be familiar with individuals who demonstrate idiosyncratic behaviours (usually when interacting with humans). Individual variation in behavioural repertoires is therefore quite crucial to establishing a really watertight behavioural baseline for a study like this, and should be acknowledged as a limitation of the design if it is not possible. The authors appear to have identified a couple of candidates for this in their ethogram ("window cleaning" and "raspberry") - it may be worth at least asking carestaff if they have observed these in all or most individuals. They could also be asked if there are any other unusual behaviours which are currently missing from the ethogram.

Secondly: The entries in the ethogram taken from Persson et al. (2017) are not useful. Currently, the authors have either interpreted their meaning independently ("There are no definitions of this behaviors in original paper, interpreted as chimpanzee makes short audible contact between knuckles and head"), or not defined them at all, both of which are inadequate. Why not contact the original authors and request proper definitions? Such a vague ethogram is not acceptable when the exact form of behaviours is so crucial to the research question at hand.

Minor comments:

The questionnaire: I think the correlation of video data with questionnaire reports on imitative

behaviour from the public will be interesting. However, I would caution against the implication that any bias identified here is likely to explain the results of Persson et al. (2017) - the

Line 281: "eight hand-reared individuals (i.e., enculturated to some degree)"

More details on what hand-rearing entailed in this context should be gathered if the authors wish to include this variable in their analysis. i.e. Were they raised in human homes, a lab, or a nursery? Were they raised alongside other chimpanzees? How many hours of human contact did they receive per day? How long were they hand-reared for? Lumping all hand-reared individuals (which could mean almost anything from 'grew up in a human household and was dressed up like a sailor' to 'was bottle-fed by a keeper twice per day in a chimpanzee nursery') in the same category as truly enculturated apes (Kanzi, Ai, Lucy, etc.) is unlikely to be informative.

Line 310: "A potential imitative event will be considered as each case when a chimpanzee and a visitor interact through a glass window or the mesh of the outdoor enclosure."

How will an 'interaction' be defined?

Line 323: "If the demonstrated action has an environmental effect (e.g. creates a sound) or not."

Just as a consideration: Zoo enclosure glass is generally very thick and may block out anything less noisy than a chimpanzee display or pant hoot.

Line 328: "From the video recordings compiled during the visitor experiment, the presence of action matching and the initiating species will be coded from each potentially imitative event a second time in order to determine if the experimenter was biased regarding the perception of imitative events, particularly the species that initiates such events."

Will the experimenter and the individual doing the second coding be the same person? What will happen if the two coders disagree with each other?

Will interactions not coded as 'potentially imitative' by the first coder also be second-coded in this fashion? If not, why not? Giving positive data points two possible opportunities for rejection and negative data points only one is likely to quickly bias the dataset.

Line 440: "We do not expect imitation probability to be affected by age, trial number, or demonstration number, nor do we expect it to vary considerably between individuals"

On the contrary, I would predict large amounts of individual variation according to how human-oriented individuals are. The authors also note elsewhere that they expect that rearing history may cause individual differences in imitative behaviour.

Lines 482-508: These model descriptions are difficult to follow. A table of all model structures used would be much easier to parse.

Lines 707-713: This is good - But what precautions are being taken to avoid false negatives?

Supplemental Material: Reproducibility is contingent on being able to understand what was being done at each step of an analysis, and why. I did not find that I could do either with this script, and feel confident most naive readers would similarly struggle. I recommend it be tidied and appropriately commented.

Author's Response to Decision Letter for (RSOS-200228.R0)

See Appendix A.

Decision letter (RSOS-200228.R1)

25-Mar-2020

Dear Ms Motes Rodrigo

On behalf of the Editor, I am pleased to inform you that your Manuscript RSOS-200228.R1 entitled "Evaluating the influence of action- and subject-specific factors on chimpanzee action copying" has been accepted in principle for publication in Royal Society Open Science.

You may now progress to Stage 2 and complete the study as approved. Before commencing data collection we ask that you:

- 1) Update the journal office as to the anticipated completion date of your study. We fully appreciate that the COVID-19 pandemic is likely to delay the onset of your research and that under the current circumstances you may be unable to even anticipate a start date, let alone a completion date.
- 2) Register your approved protocol on the Open Science Framework (<https://osf.io/>) or other recognised repository, either publicly or privately under embargo until submission of the Stage 2 manuscript. Please note that a time-stamped, independent registration of the protocol is mandatory under journal policy, and manuscripts that do not conform to this requirement cannot be considered at Stage 2. The protocol should be registered unchanged from its current approved state, with the time-stamp preceding implementation of the approved study design.

Following completion of your study, we invite you to resubmit your paper for peer review as a Stage 2 Registered Report. Please note that your manuscript can still be rejected for publication at Stage 2 if the Editors consider any of the following conditions to be met:

- The results were unable to test the authors' proposed hypotheses by failing to meet the approved outcome-neutral criteria.
- The authors altered the Introduction, rationale, or hypotheses, as approved in the Stage 1 submission.
- The authors failed to adhere closely to the registered experimental procedures. Please note that any deviations from the approved experimental procedures must be communicated to the editor immediately for approval, and prior to the completion of data collection. Failure to do so can result in revocation of in-principle acceptance and rejection at Stage 2 (see complete guidelines for further information).
- Any post-hoc (unregistered) analyses were either unjustified, insufficiently caveated, or overly dominant in shaping the authors' conclusions.
- The authors' conclusions were not justified given the data obtained.

We encourage you to read the complete guidelines for authors concerning Stage 2 submissions at <https://royalsocietypublishing.org/rsos/registered-reports#ReviewerGuideRegRep>. Please especially note the requirements for data sharing, reporting the URL of the independently

registered protocol, and that withdrawing your manuscript will result in publication of a Withdrawn Registration.

Please note that Royal Society Open Science will introduce article processing charges for all new submissions received from 1 January 2018. Registered Reports submitted and accepted after this date will ONLY be subject to a charge if they subsequently progress to and are accepted as Stage 2 Registered Reports. If your manuscript is submitted and accepted for publication after 1 January 2018 (i.e. as a full Stage 2 Registered Report), you will be asked to pay the article processing charge, unless you request a waiver and this is approved by Royal Society Publishing. You can find out more about the charges at <https://royalsocietypublishing.org/rsos/charges>. Should you have any queries, please contact openscience@royalsociety.org.

Once again, thank you for submitting your manuscript to Royal Society Open Science and we look forward to receiving your Stage 2 submission. If you have any questions at all, please do not hesitate to get in touch. We look forward to hearing from you shortly with the anticipated submission date for your stage two manuscript.

on behalf of Professor Chris Chambers (Registered Reports Editor, Royal Society Open Science)
openscience@royalsociety.org

Author's Response to Decision Letter for (RSOS-200228.R1)

See Appendix B.

RSOS-200228.R2 (Revision)

Review form: Reviewer 2

Is the manuscript scientifically sound in its present form?

No

Are the interpretations and conclusions justified by the results?

No

Is the language acceptable?

Yes

Do you have any ethical concerns with this paper?

No

Have you any concerns about statistical analyses in this paper?

No

Recommendation?

Major revision

Comments to the Author(s)

I reviewed this report in its initial stage and am happy to see it again in its finished format, which is mostly a very nicely written manuscript. Unfortunately, while I found the 'demonstration experiment' very well-implemented, the 'viewing experiment' was disappointing and I am not sure what can be done with the small quantity of data collected there. I hope that, based on my comments in the attached .pdf (Appendix C), the authors will find a way to use this data productively.

Below I provide a brief answers to the questions explicitly asked by RSOS in the review portal:

- Whether the data are able to test the authors' proposed hypotheses by passing the approved outcome-neutral criteria (such as absence of floor and ceiling effects or success of positive controls)

I do not find that the authors have sufficient data from their first experiment to test the corresponding hypotheses. I provide details on why this is the case in the attached .pdf

- Whether the Introduction, rationale and stated hypotheses are the same as the approved Stage 1 submission

The authors adhered to their original goals and hypotheses.

- Whether the authors adhered precisely to the registered experimental procedures

They did.

- Where applicable, whether any unregistered exploratory statistical analyses are justified, methodologically sound, and informative

Not applicable.

- Whether the authors' conclusions are justified given the data

Only partially. The conclusions based on their demonstration experiment are justified, but I am unconvinced by the arguments based on the data of their viewing experiment. I provide an in-depth discussion of my issues with this data in the attached .pdf.

Decision letter (RSOS-200228.R2)

Dear Ms Motes Rodrigo:

On behalf of the Editor, I am pleased to inform you that your Stage 2 Registered Report RSOS-200228.R2 entitled "Evaluating the influence of action- and subject-specific factors on chimpanzee action copying" has been deemed suitable for publication in Royal Society Open Science subject to minor revision in accordance with the referee suggestions. Please find the referees' comments at the end of this email.

The reviewers and Subject Editor have recommended publication, but also suggest some minor revisions to your manuscript. Therefore, I invite you to respond to the comments and revise your manuscript.

Please also ensure that all the below editorial sections are included where appropriate -- if any section is not applicable to your manuscript, please can we ask you to nevertheless include the heading, but explicitly state that the heading is inapplicable. An example of these sections is attached with this email.

- Ethics statement

- Data accessibility

If you wish to submit your supporting data or code to Dryad (<http://datadryad.org/>), or modify your current submission to dryad, please use the following link:
[http://datadryad.org/submit?journalID=RSOS&manu=\(Document not available\)](http://datadryad.org/submit?journalID=RSOS&manu=(Document not available))

- Competing interests

- Authors' contributions

AB carried out the molecular lab work, participated in data analysis, carried out sequence alignments, participated in the design of the study and drafted the manuscript; CD carried out

the statistical analyses; EF collected field data; GH conceived of the study, designed the study, coordinated the study and helped draft the manuscript. All authors gave final approval for publication.

- Acknowledgements

- Funding statement

Because the schedule for publication is very tight, it is a condition of publication that you submit the revised version of your manuscript within 7 days (i.e. by the 13-Jan-2021). If you do not think you will be able to meet this date please let me know immediately.

on behalf of Professor Chris Chambers
 (Registered Reports Editor, Royal Society Open Science)
 openscience@royalsociety.org

Associate Editor Comments to Author (Professor Chris Chambers):

Associate Editor: 1

Comments to the Author:

One of the expert reviewers from Stage 1 was available to assess the Stage 2 submission. As you will see, the reviewer offers a number of constructive suggestions for revision. In revising, please note that the approved study design is not relitigated at Stage 2, but any limitations raised should nevertheless be addressed in the Discussion. Please also avoid making any changes to the approved Stage 1 part of the manuscript that are not necessary to correct factual errors or avoid confusion.

Comments to Author:

Reviewer: 2

Comments to the Author(s)

I reviewed this report in its initial stage and am happy to see it again in its finished format, which is mostly a very nicely written manuscript. Unfortunately, while I found the 'demonstration experiment' very well-implemented, the 'viewing experiment' was disappointing and I am not sure what can be done with the small quantity of data collected there. I hope that, based on my comments in the attached .pdf, the authors will find a way to use this data productively.

Below I provide a brief answers to the questions explicitly asked by RSOS in the review portal:

- Whether the data are able to test the authors' proposed hypotheses by passing the approved outcome-neutral criteria (such as absence of floor and ceiling effects or success of positive controls)

I do not find that the authors have sufficient data from their first experiment to test the corresponding hypotheses. I provide details on why this is the case in the attached .pdf

- Whether the Introduction, rationale and stated hypotheses are the same as the approved Stage 1 submission

The authors adhered to their original goals and hypotheses.

- Whether the authors adhered precisely to the registered experimental procedures

They did.

- Where applicable, whether any unregistered exploratory statistical analyses are justified, methodologically sound, and informative

Not applicable.

- Whether the authors' conclusions are justified given the data

Only partially. The conclusions based on their demonstration experiment are justified, but I am unconvinced by the arguments based on the data of their viewing experiment. I provide an in-depth discussion of my issues with this data in the attached .pdf.

Author's Response to Decision Letter for (RSOS-200228.R2)

See Appendix D.

Decision letter (RSOS-200228.R3)

Dear Ms Motes Rodrigo:

It is a pleasure to accept your revised Stage 2 Registered Report entitled "Evaluating the influence of action- and subject-specific factors on chimpanzee action copying" in its current form for publication in Royal Society Open Science.

on behalf of Professor Chris Chambers (Subject Editor)
openscience@royalsociety.org

Appendix A

Comments to the Author:

Two expert reviewers have now assessed the manuscript. Both find significant merit in the proposal while also pointing to a range of issues that will be need to addressed to achieve in principle acceptance, from questions of feasibility (in particular, expectations of visitor observation time - a concern highlighted by both reviewers), methodological detail, sample planning, and presentational clarity. All issues are within the range of concerns that are generally addressable for a Stage 1 RR therefore a Major Revision is invited. Please respond carefully and comprehensively to each point in the reviews.

We thank the reviewer's for their insightful comments that have significantly improved the clarity of the manuscript. We address each of the reviewer's comments below.

Comments to Author:

Reviewer: 1

Comments to the Author(s)

Please find my comments regarding each of the key aspects of the proposal below.

• The scientific validity of the research question(s)

The authors outline a planned study of imitation in captive chimpanzees, using two experimental approaches. The first will attempt to replicate (with refinements to the methodology) results from Persson et al., who report imitation between chimpanzees and zoo visitors. The second experiment will investigate the potential factors which contribute to imitative abilities in chimpanzees (such as enculturation, familiarity of the action), using individual testing of captive chimpanzees.

The questions this study aims to address are clearly explained and are scientifically valid. Replicating the Persson et al. study is a good idea (with the proposed addition of video recording to strengthen the methodology). The second experiment would be a useful addition to the literature by considering together multiple factors impacting imitation.

We thank the reviewer for supporting this project. We have addressed each of the reviewer's comments below.

1. The logic, rationale, and plausibility of the proposed hypotheses

It might be useful if the authors could add explicit hypotheses alongside the aims of their studies. Their data predictions appear to me to be well reasoned and clear.

Following the reviewer's suggestion we have now stated our hypothesis alongside our aims in lines 165-167 and lines 180-183.

2. The soundness and feasibility of the methodology and analysis pipeline (including statistical power analysis where applicable)

2.1 Visitor experiment:

You mention a planned sample size of 25 participants for ~15 hours of interactions. This implies around 30 minutes of interaction time per participant, and given you won't impose a minimum interaction time, I think that may be a little long to expect people to stay for?

I would consider clarifying in your planned methods whether your goal in terms of the sample size for this study is the number of participants, or the number of hours of interaction time - my suspicion is you may need to recruit more than 25 participants to reach 15 hours of interactions. I would also suggest planning to recruit a larger sample of participants as you may find substantial inter-individual variation in the behaviours the human participants attempt, and this would also give you a larger sample for your interesting question regarding humans' potential bias towards perceiving imitation.

We agree with the reviewer that it would be beneficial to increase the number of visitors tested during the Visitor condition in order to compensate for potentially short interactions times and to

ensure that we increase inter-individual variation in the perception of imitation. Consequently we have modified our stopping point at 50 visitors or 20 hours of interaction in lines 264-265.

2.2. Demonstration experiment:

The choice of actions seems sensible to me, and the choice of both contact / non-contact actions, and actions with and without environmental effects is good.

How will you select the 8 mother-reared individuals? Will they be age and sex matched (as far as possible) with the hand-reared individuals? Or the individuals considered most likely to engage in the testing process? Or randomly selected?

As the reviewer suggests, we will try to match the sex ratio (3:5) and the mean age (28) of the hand-reared individuals with the mother-reared individuals tested in the Demonstration experiment. However, as the participation of the chimpanzees in the study will be voluntary, it is possible that we will end up testing the more engaged individuals (those more comfortable with entering the off-sight quarters). We have added more information in this regard in lines 302-306.

The coding schemes for both experiments seem well-planned and thorough. I think there's a typo in the Demonstration coding scheme, it reads "the chimpanzees or the humans perform an action after an action has been demonstrated." In the Demonstration experiment, only the chimpanzees could imitate, correct? The human demonstrators presumably will be instructed not to imitate the chimpanzees in this experiment?

We have now corrected this typo in line 390 and we have also specified that the human demonstrators will be instructed not to imitate the chimpanzees in the Demonstration experiment in lines 402-403.

2.3 You may wish to also code for the identity of the demonstrator in these trials, if there's the possibility of using different demonstrators. Some chimpanzees may have closer relationships with some keepers than others, and I can imagine this could impact their level of attention to the demonstrations at least.

We agree with the reviewer that the identity of the demonstrator might influence the level of attention of some chimpanzees. Consequently, we will try that only one keeper performs all demonstrations in order to control for demonstrator identity. However, if due to unforeseen circumstances, we need to use multiple demonstrators, their identity will be coded as suggested by the reviewer. We have specified this in lines 269-272.

2.4 The power analysis / simulation is very convincing. It isn't clear to me from the current report what your approach will be if you find more than 0, but fewer than 46 imitation events in your data set – you state you will only fit your full model with 46 events, in order to have good power. Would you attempt any alternative statistical analysis with fewer than 46 events? Would you be able to draw any conclusions from fewer than 46 events – from your power analysis I understand it would not be possible to explore the factors contributing to imitation – how will you then present the results?

The results of our analyses of simulated data revealed that with too few imitation events we cannot fit a model allowing for a reliable estimation of the contribution of the investigated factors to the probability of an action to be imitated. Hence, in such a case we shall not fit a model, but just describe the occurrence of imitation events qualitatively. The results, however, could be used to inform a follow up study aiming at a sample size sufficiently large enough to fit an adequate model.

3. Whether the clarity and degree of methodological detail would be sufficient to replicate exactly the proposed experimental procedures and analysis pipeline

If the above comments are addressed, I believe the methods and analysis could be replicated by others.

We thank the reviewer for this positive feedback.

4. Whether the authors provide a sufficiently clear and detailed description of the methods to prevent undisclosed flexibility in the experimental procedures or analysis pipeline
I think the Visitor experiment needs a little clarification regarding the ‘stopping point’, whether this will be the number of participants or hours of interaction time, but aside from this I think the methodology for this study is clear.

We have clarified the stopping point of data collection in line 264-265: "Data collection will continue until 50 zoo visitors have participated in the study or 20 hours of video recordings have been collected."

5. Whether the authors have considered sufficient outcome-neutral conditions (e.g. positive controls) for ensuring that the results obtained are able to test the stated hypotheses
The study seems well planned. The planned use of double-coding from video, with paired dummy examples, seems a sufficient quality control.

We thank the reviewer for this positive feedback. Based on the comments by Reviewer 2 we have modified the coding scheme so the second coder will code all events where a chimpanzee and a visitor were within 2 m through the glass window for the presence or absence of imitation. These events will include the interactions that the experimenter (first coder) coded as containing imitation as well as those where the first coder did not detect imitation (lines 352-355).

Additional comments -

Participant info sheet / consent forms

6. Typo in final paragraph of info sheet (switches from ‘you will’ to ‘my name will never appear’)

We have now corrected this typo.

7. This may be a non-issue following translation into German, but I would suggest giving a very brief explanation of what imitation is using lay terms (e.g. “we’re interested in seeing whether the apes will copy you”). I would consider ‘imitate’ a somewhat technical term in English and would simplify it for zoo visitors – but perhaps this won’t be a problem once the document is translated.

We have added this information in the "General information for participants" form.

8. I would consider adding something to the info sheet to discourage the potential bad behaviours you list in your methods – I think it’s good that you plan to end the trial if a participant bangs loudly on the glass, for example, but it might be even better to explicitly discourage this behaviour before you start testing. I don’t think this would place too great a limitation on the behaviours participants perform when trying to interacting with the apes.

We have included the following sentence in the "General information for participants" form: "We discourage any behavior that might disturb the apes such as loudly banging on the glass. If inappropriate behaviors take place, we will stop the experiment."

Reviewer: 2

Comments to the Author(s)

Overall, I found this to be a clear and robust proposal for an important piece of work. The authors propose a replication of a contentious study by Persson et al. (2017) studying between-species imitation of chimpanzees and humans, as well as exploring interesting additional questions regarding human bias towards perception of imitative events. Crucially, the authors propose considerable methodological refinements to the methods used by Persson et al. through the use of video recordings and multiple coders as well as creating

a 'behavioural baseline' for their chimpanzee sample, to ensure that any seemingly imitative behaviours are indeed novel.

A particular strength of the proposal is that the authors' plan to take a 'two prong' approach, using side-by-side observational and experimental designs. The second, experimental portion of the study will test for human-directed imitation in chimpanzees under a more controlled setting. The authors are interested in exploring the various individual and action-related factors that influence the likelihood with which imitation events occur. The authors have approached this portion of the study with a level of rigour and preparation which is rare in the field.

We thank the reviewer for the kind words and for supporting this project. Please find our answers to the reviewer's comments below.

I have three main issues which the authors should address:

- 1) While the authors carried out careful power calculations for the demonstration experiment, I have major concerns regarding the proposed sampling size and logistics for the observational study.**
- 2) The use of 'finger pop' as the candidate for imitative behaviour in the demonstration experiment is problematic.**
- 3) There are two potentially significant but easily resolved issues regarding the behavioural ethogram proposed for determining a behavioural baseline for the chimpanzees. Firstly, a denial of the significance of individual differences in behavioural repertoires for a study of this kind and secondly a lack of clear definition on many of the items in their ethogram.**

I have detailed each of these major comments below, as well as a handful of more minor comments.

We thank the reviewer for the feedback and respond to each of the points below.

I would emphasise that each of the issues I have raised would simply require refinement of the proposed methods rather than any sort of overhaul, and would not anticipate that they be barriers to completion of the work or its subsequent publication.

Major comments:

Sampling

In my experience of zoos, it seems unlikely that visitors will spend, on average, 30+ minutes observing the chimpanzees. The public tends to stick around for long periods during events such as feeding or when the animals are otherwise particularly active (and therefore perhaps less likely to interact with the public), but during rest periods may only spend a few minutes waiting for something interesting to happen and then leave. The researchers should therefore consider how they will choose the timing of their sampling periods, and what effects this might have in terms of the likelihood that individuals will interact with the public (e.g. during scatter feeds, I would expect zero interaction events).

We thank the reviewer for the comment. We agree with the reviewer that the activity levels of the chimpanzees and the frequency of the interactions with the visitors will vary throughout the day. Consequently, our aim is to collect data from visitor interactions throughout the day both during feeding and not-feeding times to account for this variation. Following the reviewer's suggestion we have indicated in line 350 that we will also record the time of the day at which visitors are filmed.

In light of the above the researchers may need to revise either a) their expectations for the number of hours they will sample, or b) their target number of participants. If not, they should decide whether they will stop at 25 participants regardless of how many hours of interaction data they have collected (if the researchers are ‘unlucky’ they could easily sample 25 individuals who each spent less than 5 minutes in the exhibit or are never afforded an opportunity to ‘interact’ with a chimpanzee), or continue sampling until a secondary criteria of time is met.

We agree with the reviewer that it would be beneficial to increase the number of visitors tested during the Visitor condition in order to compensate for potentially short interactions times. Consequently we have modified our stopping point at 50 visitors or 20 hours of interactions (lines 264-265).

A more specific description of how sampling will take place is also important. Will multiple individuals from a group of visitors be sampled? If so, can these be considered independent data points? If not, are we to assume that participants will be taken to a more private area, since non-consenting individuals cannot be recorded on video?

To avoid pseudoreplication, visitors will be tested individually in a cordoned area where no other visitors will be allowed to enter. In addition to prevent different participants to influence each other's behaviour, this measure will ensure that we do not film visitors that have not provided signed consent. We have specified this in lines 250-252.

The authors may wish to consider how this will influence a) how long participants will spend at the enclosure (probably not long, if their family/friends are waiting for them) and b) how this might influence chimpanzee behaviour (are they more likely to spend time at windows with more people present?).

We agree with the reviewers that this set up might reduce the time visitors spend interacting with the chimpanzees and vice versa. However, we believe that individual testing is a necessary condition in order to ensure that we avoid pseudoreplication due to visitors influencing each other's behavior when being tested together. In an attempt to compensate for short periods of interaction, we have modified our stopping point at 50 visitors or 20 hours of interactions (264-265).

While I think the use of video recordings will yield high quality data in the proposed study, it is also a limiting factor. Persson et al.'s methods were no doubt noisier, but had the advantage of sampling ALL individuals who passed through the exhibit during an extensive sampling period. To illustrate: 52 hours of observation in Persson et al. (2017) in which they were able to sample interactions between the chimps and let's say, conservatively, 10 individuals visiting at any given moment (it was during the busiest periods) - effectively yielding 520 hours of potential interaction time. They recorded 3794 interaction observations, 10% of which were reported to be imitative: a rate of 0.7 potentially imitative acts per person per hour. Assuming this figure to be accurate, and that approximate interaction rates are consistent between study sites, how many observations of interest would we expect from a replication of just 15 hours from 25 individuals? I am not convinced by the utility of such a small sample.

Following the reviewer's suggestion, we have modified our stopping point at 50 visitors or 20 hours of interactions to increase our sample size during the Visitor experiment (please see previous comment).

Our goal with this experiment is to obtain records of the interaction between the chimpanzees and the visitors that are as detailed as possible, specially regarding the timelines of the actions performed. As the reviewer mentioned in a previous comment, testing the interaction rates of groups of visitors (rather than single visitors) creates the problem that the data points can hardly be considered independent. Therefore, despite the impressive number of hours and interactions

that were included in Persson et al.'s study, it is unclear if there exists pseudoreplication in their data. We acknowledge that the set up of our Visitor experiment somehow limits the number of hours of observation that we can realistically collect and thus our experiment will yield a lower number of hours/events than those collected by Persson et al. However, our design ensures the independence of data points. In addition, the visitors included in our experiment will be instructed to try to make the apes imitate them. We suspect that these instructions will elicit an increase in the number of interactions that the visitors seek to engage in, potentially compensating for the lower number of hours included in our study compared to Persson et al.

Methods: “Finger pop sound (hook finger inside mouth and release with a sound)” as action demonstrated in second experiment.

Are there any recorded examples of a chimpanzee doing this? I’m not certain that it is physiologically possible for them to produce such a sound as it requires quite a bit of control over the articulators (lips + cheek) and airflow in order to achieve an audible ‘pop’, which are motor functions not generally thought to be under much top-down control by chimpanzees.

The opacity of the causal factors in achieving this sound also make it a poor candidate for examining imitation. Indeed, the authors include a link to a ‘wiki how’ article providing instructions on how to perform a finger-pop and a quick google search brings up a list of similar articles and Youtube videos (in a quick survey of my friends I found 2 in 10 could not, and another could only do so with practice). If this degree of explicit verbal instruction is necessary for some humans, it does not seem a ‘fair’ behaviour for use here. Unless the authors have a strong justification for this choice, therefore, I strongly recommend choosing another behaviour.

We thank the reviewer for taking the time to conduct a quick study on finger popping. Following the reviewer's results, we have changed this action to "strum lips" (Table 1). As mentioned in lines 383-384, imitation will be coded as present if the reproduction of the demonstrated action is at least partial. Therefore, even if the chimpanzees do not perform the action exactly as demonstrated (e.g. no audible sound is produced by lip strumming) but the body parts match the demonstration, partial imitation would still be coded. In addition, the crucial action that needs to produce a sound, involve contact and be unfamiliar to the chimpanzees is the one performed by the demonstrator.

The ethogram

Firstly: The authors state that collecting individual action repertoires is not necessary or reasonable (Footnote 1). I am surprised by this - anyone who has worked with captive chimps is likely to be familiar with individuals who demonstrate idiosyncratic behaviours (usually when interacting with humans). Individual variation in behavioural repertoires is therefore quite crucial to establishing a really watertight behavioural baseline for a study like this, and should be acknowledged as a limitation of the design if it is not possible. The authors appear to have identified a couple of candidates for this in their ethogram (“window cleaning” and “raspberry”) - it may be worth at least asking carestaff if they have observed these in all or most individuals. They could also be asked if there are any other unusual behaviours which are currently missing from the ethogram.

We agree with the reviewer that idiosyncratic behaviours need to be accounted for when compiling behavioural baselines. We realised that the wording of the footnote was misleading as we did not mean to say that idiosyncratic behaviours would not be included in the baseline. In the footnote, we meant to clarify that when compiling the baseline, we recorded all behaviours present in the population regardless of the chimpanzee' identity or the number of individuals that performed the behaviour. We have now rewritten the footnote (now number 2) following the reviewer's comments to improve its readability. We will also follow the reviewer's suggestion and before the onset of the experiments, we will individually ask each of the zookeepers if they have

seen additional behaviours in the chimpanzee group which are not included in our ethogram. We have specified this in lines 328-331.

Secondly: The entries in the ethogram taken from Persson et al. (2017) are not useful. Currently, the authors have either interpreted their meaning independently (“There are no definitions of this behaviors in original paper, interpreted as chimpanzee makes short audible contact between knuckles and head”), or not defined them at all, both of which are inadequate. Why not contact the original authors and request proper definitions? Such a vague ethogram is not acceptable when the exact form of behaviours is so crucial to the research question at hand.

Following the reviewer's comment, we contacted the corresponding author of the Persson et al. (2017) study on the 6th of March requesting complete definitions of the behaviours included in their ethogram. Unfortunately, we have not received any answer and thus, we decided to exclude these behaviours from the ethogram in our study. To account for behaviours that we might have missed during the data collection for our baseline, we will review our ethogram with the chimpanzee keepers at the zoo before starting our study in order to include any other behaviours to our list that might be currently missing (as suggested by the reviewer in the previous comment).

Minor comments:

The questionnaire: I think the correlation of video data with questionnaire reports on imitative behaviour from the public will be interesting. However, I would caution against the implication that any bias identified here is likely to explain the results of Persson et al. (2017) - the

Following the reviewer' suggestion we have removed this statement from the manuscript.

Line 281: “eight hand-reared individuals (i.e., enculturated to some degree)”

More details on what hand-rearing entailed in this context should be gathered if the authors wish to include this variable in their analysis. i.e. Were they raised in human homes, a lab, or a nursery? Were they raised alongside other chimpanzees? How many hours of human contact did they receive per day? How long were they hand-reared for? Lumping all hand-reared individuals (which could mean almost anything from ‘grew up in a human household and was dressed up like a sailor’ to ‘was bottle-fed by a keeper twice per day in a chimpanzee nursery’) in the same category as truly enculturated apes (Kanzi, Ai, Lucy, etc.) is unlikely to be informative.

Following the reviewer's comment we have included more details on what we mean in the manuscript by human-reared and the background of these chimpanzees in lines 295-302. In this manuscript, we refer to human-reared chimpanzees as those individuals that during their first year of life had extensive human contact, namely they lived for a certain period of time in human homes (the exact length is not possible to determine as many of the individuals were abandoned at the zoo entrance by the previous private keepers) or they lived in a nursery group of conspecifics at the zoo but were bottle fed every day because their mothers rejected them. Although we agree that different degrees of human exposure and the timing of this exposure can lead to behavioural differences during adulthood, all the human-reared individuals in our sample have in common that during their first year of live they were taken care of by the zookeepers (in nursery groups or human homes). On the other hand, the mother-reared individuals in our sample lived in conspecific groups with individuals of different age classes and were reared by their mothers.

Line 310: “A potential imitative event will be considered as each case when a chimpanzee and a visitor interact through a glass window or the mesh of the outdoor enclosure.”

How will an ‘interaction’ be defined?

We have clarified this is in lines 334-335. A potential imitative event will take place when the visitors and the chimpanzees remain face to face within 2 m of each other. From all potential imitative events the presence or absence of imitation will be coded first live by the experimenter and then from video recordings by a second coder.

Line 323: “If the demonstrated action has an environmental effect (e.g. creates a sound) or not.”

Just as a consideration: Zoo enclosure glass is generally very thick and may block out anything less noisy than a chimpanzee display or pant hoot.

We thank the reviewer for the comment. In Leintal zoo, only some sections of the enclosure walls are covered with glass, whereas most of the enclosure is surrounded by mesh (even around the glass windows). Therefore, we believe (also based on our previous experience at the zoo) that most sounds produced by the chimpanzees will be audible from the visitor area.

Line 328: “From the video recordings compiled during the visitor experiment, the presence of action matching and the initiating species will be coded from each potentially imitative event a second time in order to determine if the experimenter was biased regarding the perception of imitative events, particularly the species that initiates such events.”

Will the experimenter and the individual doing the second coding be the same person? What will happen if the two coders disagree with each other?

The second coder will be a different person from the experiment (352-355). If the two coders don't agree, the data point will be marked as not reliable and reported as "ambiguous" (427-428).

Will interactions not coded as ‘potentially imitative’ by the first coder also be second-coded in this fashion? If not, why not? Giving positive data points two possible opportunities for rejection and negative data points only one is likely to quickly bias the dataset.

Following the reviewer's comments we have now clarified that potentially imitative interactions will be those in which the visitor and the chimpanzee are within 2 m of each other. The second coder will code all potentially imitative events for the presence of imitation, regardless if the first coder perceived imitation or not (lines 354-355).

Line 440: “We do not expect imitation probability to be affected by age, trial number, or demonstration number, nor do we expect it to vary considerably between individuals”

On the contrary, I would predict large amounts of individual variation according to how human-oriented individuals are. The authors also note elsewhere that they expect that rearing history may cause individual differences in imitative behaviour.

We agree with the reviewer that there will be differences in performance between individuals, which is why we included this variable in the models as a random slope (see Table 2 and lines 523-524). We have rewritten the sentence mentioned by the reviewer to improve clarity.

Lines 482-508: These model descriptions are difficult to follow. A table of all model structures used would be much easier to parse.

Following the reviewer's comment we have now created a table (Table 4) with all the model structures used in the analysis.

Lines 707-713: This is good - But what precautions are being taken to avoid false negatives?

As described in a previous comment and following the reviewer's suggestions, all potentially imitative interactions where the chimpanzees are within 2 m of the visitors will be recoded by a second coder from video recordings. Only if both coders agree that an interaction involves an imitative event, this will be considered as such. If the coders disagree, the event will be reported as ambiguous (428-429).

Supplemental Material: Reproducibility is contingent on being able to understand what was being done at each step of an analysis, and why. I did not find that I could do either with this script, and feel confident most naive readers would similarly struggle. I recommend it be tidied and appropriately commented.

We have now included detailed comments in the script uploaded to the OSF as well as a file with the necessary functions to run the code.

Appendix B

EBERHARD KARLS
UNIVERSITÄT
TÜBINGEN

Faculty of Mathematics
and Natural Sciences

To the Editorial Board of the Royal Society Open Science

Department of Early Prehistory and
Quaternary Ecology

Alba Motes Rodrigo
Burgsteige 11
72070 Tübingen
Germany
alba.motes-rodrigo@uni-tuebingen.de
albamotes7@gmail.com

Dear Editor,

We are writing with regard to our manuscript "**Evaluating the influence of action- and subject-specific factors on chimpanzee action copying**" which received In Principle Acceptance (IPA) in Royal Society Open Science on March 2020 (manuscript ID RSOS-200228.R1). We are pleased to inform the Editorial Board that we completed data collection this summer and therefore, we are now submitting our Stage 2 manuscript.

Following the Instructions for Authors of Stage 2 Registered Reports submissions, we confirm that our manuscript contains in page 27 the URL of the OSF folder where our raw data, R code and approved Stage 1 protocol are publicly available. We further confirm that no data was collected prior to the date of the IPA.

Kind regards,

Alba Motes Rodrigo, Roger Mundry, Josep Call and Claudio Tennie

Appendix C

Please see below for my in-depth comments on this manuscript. I have divided this into three sections corresponding to the two experiments and power analysis of the manuscript. Minor comments and typos are listed at the end of this document.

Viewing Experiment

In observational work there is always a tension between collecting high-quality data, and collecting a sufficient quantity of data. Doing both is time consuming, and one is not a shortcut around the other. Here, the authors have collected very high fidelity data, but very little of it. Sadly, I found that there is simply not enough data here to draw any inference from, much less compare with the work of Persson et al., whose study the authors aim to replicate, refine and ultimately rebuke.

This was highlighted as a major concern in my original review of the report, as follows:

“While I think the use of video recordings will yield high quality data in the proposed study, it is also a limiting factor. Persson et al.’s methods were no doubt noisier, but had the advantage of sampling ALL individuals who passed through the exhibit during an extensive sampling period.

To illustrate: 52 hours of observation in Persson et al. (2017) in which they were able to sample interactions between the chimps and let’s say, conservatively, 10 individuals visiting at any given moment (it was during the busiest periods) - effectively yielding 520 hours of potential interaction time. They recorded 3794 interaction observations, 10% of which were reported to be imitative: a rate of 0.7 potentially imitative acts per person per hour.

Assuming this figure to be accurate, and that approximate interaction rates are consistent between study sites, how many observations of interest would we expect from a replication of just 15 hours from 25 individuals? I am not convinced by the utility of such a small sample.”

While the authors adjusted their planned sample somewhat (from 25 to 50 individuals), I regret that was I not given another opportunity to review the proposal after their revisions, as I would have emphasised that even this revised sample was still likely to be inadequate. Indeed, the number of participants is relatively unimportant – it is the amount of time actually spent recording behaviors that is of importance.

To illustrate my issue more specifically: The authors state that the zoo guests (N = 50) participated for 5h30 in total, but as far as I can tell they do not directly state for how much of this time was a chimpanzee within 2m (i.e. their actual data collection window). I can only infer the true amount from the the following on Line 786.

Line 786: “The average duration of the visitors' engagement when potential imitative events took place was 14 min and 4 seconds (duration range 1 minute and 57 seconds to 45 minutes and 12 seconds).”

Multiplying this average out by the number of participants for whom “potential imitative events” (chimpanzee approached within 2m) actually occurred (N = 9) would indicate that the authors only actually collected ~2h of actual data, from 9 human subjects and an unknown number of chimpanzees. This is, at best (i.e. not considering the fact that Persson et al. observed groups of humans), less than 5% of the observational data collected (52h) in the study the authors claim to be replicating.

Line 934: Persson et al. (2017) only tested five chimpanzees, four of which had been at least partly human-reared. In our study, we tested 32 chimpanzees, only seven of which had been partly human-reared”

Given the issues described above, statements like this feel rather disingenuous. How many of these 32 chimpanzees actually came within 2m of the human participants (the author’s criteria for inclusion)? Given that only 30 potential imitative events occurred, it cannot be all of them. Any who did not do so were not “tested” and should be removed from this appeal to superior sample size. The authors did not observe these individuals.

To be clear, chimpanzees approaching humans in close proximity is not in itself a behaviour of interest: The authors self-imposed a criteria of chimps being within 2m for ease of coding, not because imitative events can only occur, or are even necessarily most likely to occur, within this range. The term ‘potential imitative event’ is a poor fit then, since imitative events could occur at any number of other potential ranges, which the authors exclude by implication with this term. Moreover, the fact that close-approaches were rare does not, therefore, present evidence either way for imitative behaviour in chimpanzees, although the authors make it sound so in the Discussion:

“We found that potential imitative events (when the chimpanzees and visitors were less than 2m apart through the glass window) were very rare. Even though we instructed the zoo visitors to try to make the chimpanzees imitate them (and consequently the visitors gestured and performed a wide variety of actions), the chimpanzees hardly ever came in close proximity with the visitors.”

This lab is known for its (well-justified!) claims that primatology has a bias towards methods and mindsets which wrongly claim the existence of imitation and other ‘human-like’ behaviour in apes. I am sympathetic to this! However, it cuts both ways. It is hard to take seriously the disparity between the effort put into collecting the ‘behavioural baseline’ (forming the null hypothesis in this study (38 hours of observation, keeper surveys, etc) and the amount of actual data available to test it (2h of observation).

In light of this limitation, great caution should be taken in making any claims based on this data, and any inference drawn from them that Persson et al. “failed to replicate” or “does not generalise” should be removed from the manuscript. This was not a fair replication of their work.

As a final methodological consideration: Is it possible that people being singly observed and recorded on video by researchers are more inhibited with regards to the actions they perform

in front of the chimpanzees, either in terms of frequency or exaggeration of form? This seems likely to me, and places yet further importance on collecting a substantial amount of data to offset the effect.

Power Analysis

I commend the thorough power analysis conducted here, but I also question whether all 13 pages, 8 figures and 3 tables are conducive to comprehension and readability of the final manuscript (particularly since it does not refer to the actual sample used, see below). I suggest the authors at least consider breaking this up into further sub-sections that make it easier to follow, or (my preference) consider relegate some of this information or visualisations to supplemental material.

Line 967: “Using a simulation we estimated that in order to address our second aim with enough statistical power we needed to identify at least 46 imitation events. Although this number would need to be slightly adjusted due to the death of a chimpanzee included in our original estimation...”

Why not just re-run the simulations using the actual sample size (N = 14, not 16) of chimpanzees? Maybe the authors are correct in thinking it is won't make a difference, but we should not have to take their word on it.

Demonstration experiment

No issues for me here, this seems a very robust experiment and the authors are even-handed about its outcomes. Well done!

Minor comments and typological corrections

The abstract is rather vague. In present state, a reader knows a study has been replicated, but has no idea which one, nor are any of the current manuscript's methods described.

Line 58: “demonstrated to the two chimpanzees 48 actions” should be “demonstrated 48 actions to the two chimpanzees”

Line 84: ‘Contextual imitation’ needs to be more clearly defined on its first usage.

Line 136: ‘et al’ needs a full stop after ‘al’

143-148: How does this fourth thing differ from the ‘baseline repertoire’ referenced as the first thing on line 136?

Line 183: It’s not clear here whether the power analysis is relevant to both observation and demonstration experiments, or just the latter. It is also not clear what the ‘average’ being referred to is – averaged across what?

Appendix D

Dear Royal Society Open Science Editorial Office,

We thank the editor and reviewer for their evaluation of our manuscript (ID RSOS-200228.R1). We have now responded to all of the reviewer's comments and have edited the manuscript in the light of the suggestions made by the reviewer. Given that some of the comments of the reviewer referred to the Introduction and Methods section of our manuscript, we have made minimal changes to improve clarity and avoid misunderstandings in this sections. Following also the reviewer's comment, we have moved two figures and one table from the section of the manuscript where we described and reported our power simulation to the supplementary. Please find our detailed responses to the reviewer's comments and a detailed account of the changes we performed in the manuscript below.

Thank you very much for your time and consideration,

Sincerely,

Alba Motes Rodrigo, Roger Mundry, Josep Call and Claudio Tennie

Reviewer: 2

Comments to the Author(s)

I reviewed this report in its initial stage and am happy to see it again in its finished format, which is mostly a very nicely written manuscript.

Unfortunately, while I found the 'demonstration experiment' very well-implemented, the 'viewing experiment' was disappointing and I am not sure what can be done with the small quantity of data collected there. I hope that, based on my comments in the attached .pdf, the authors will find a way to use this data productively.

—> **RESPONSE:** We thank the reviewer for the thorough examination of our manuscript and for the valuable comments and suggestions provided. Below, we answer in blue to each of the points raised by the reviewer and report (when applicable) the corresponding changes that we have done in the manuscript.

Below I provide a brief answers to the questions explicitly asked by RSOS in the review portal:

- Whether the data are able to test the authors' proposed hypotheses by passing the approved outcome-neutral criteria (such as absence of floor and ceiling effects or success of positive controls)

I do not find that the authors have sufficient data from their first experiment to test the corresponding hypotheses. I provide details on why this is the case in the attached .pdf

—> **RESPONSE:** We have addressed in detail the reviewer's criticism below. We agree with the reviewer that longer participation times by each zoo visitor would have been desirable and that our sample size for the "Visitor experiment" is smaller than that compiled by Persson et al. (2017). However, given the method of data collection implemented by Persson et al. (observations of groups of visitors that led to pseudoreplication), their lack of video recordings and their lack of reliability analysis, we consider that it is not possible to assess the real size of the data set collected by Persson et al. Moreover, the nature or Registered Reports

is built on the premise that the methods are judged prior to the results. Our methods were reviewed and approved before data collection started and our data collection method was implemented exactly as in the Stage 1 report. Therefore, although we agree that larger sample sizes are generally beneficial, at this stage we cannot increase the testing times. Nevertheless, acknowledging the criticisms made by the reviewer, we have been more cautious in how we phrase some of our conclusions.

- Whether the Introduction, rationale and stated hypotheses are the same as the approved Stage 1 submission

The authors adhered to their original goals and hypotheses.

- Whether the authors adhered precisely to the registered experimental procedures

They did.

- Where applicable, whether any unregistered exploratory statistical analyses are justified, methodologically sound, and informative

Not applicable.

- Whether the authors' conclusions are justified given the data

Only partially. The conclusions based on their demonstration experiment are justified, but I am unconvinced by the arguments based on the data of their viewing experiment. I provide an in-depth discussion of my issues with this data in the attached .pdf.

—> RESPONSE: We have addressed some of the limitations pointed out by the reviewer in lines 977 and 996-1002 of the discussion.

(REVIEWER COMMENTS PROVIDED IN ATTACHED PDF START HERE)

Viewing Experiment

In observational work there is always a tension between collecting high-quality data, and collecting a sufficient quantity of data. Doing both is time consuming, and one is not a short-cut around the other. Here, the authors have collected very high fidelity data, but very little of it. Sadly, I found that there is simply not enough data here to draw any inference from, much less compare with the work of Persson et al., whose study the authors aim to replicate, refine and ultimately rebuke.

This was highlighted as a major concern in my original review of the report, as follows:

“While I think the use of video recordings will yield high quality data in the proposed study, it is also a limiting factor. Persson et al.’s methods were no doubt noisier, but had the advantage of sampling ALL individuals who passed through the exhibit during an extensive sampling period.

To illustrate: 52 hours of observation in Persson et al. (2017) in which they were able to sample interactions between the chimps and let’s say, conservatively, 10 individuals visiting at any given moment (it was during the busiest periods) - effectively yielding 520 hours of potential interaction time. They recorded 3794 interaction observations, 10% of which were reported to be imitative: a

rate of 0.7 potentially imitative acts per person per hour.

Assuming this figure to be accurate, and that approximate interaction rates are consistent between study sites, how many observations of interest would we expect from a replication of just 15 hours from 25 individuals? I am not convinced by the utility of such a small sample.”

While the authors adjusted their planned sample somewhat (from 25 to 50 individuals), I regret that was I not given another opportunity to review the proposal after their revisions, as I would have emphasised that even this revised sample was still likely to be inadequate. Indeed, the number of participants is relatively unimportant – it is the amount of time actually spent recording behaviors that is of importance.

—> **RESPONSE:** We thank the reviewer for this detailed evaluation of our experimental design and for the constructive criticism. We agree with the reviewer that given our goal of collecting as rigorous data as possible, our final data set was unavoidably smaller than the one presented by Persson et al. (2017). However, it is important to note that it is not possible to assess the true size of the data set compiled by Persson et al. (2017) because their data points were not independent. Consequently, the data collected by Persson et al. (2017) was likely confounded by pseureplication (please see below our comment on individual versus group testing). Furthermore, no adequate inter-rater reliability was conducted on the data collected as no video recordings were made that could have been coded by independent coders. These methodological constraints limit the conclusions that can be drawn from the study by Persson et al. (2017).

The reviewer mentions that we should have focused on attaining a larger "time actually spent recording behaviours" rather than recruiting a high number of participants. Although we agree that longer participation times by the zoo visitors would have potentially yielded a larger data set, we do not believe that prioritizing time over visitor variability would have been a more fruitful study design. Given the low amount of time (23 minutes over 50 visitors) that the chimpanzees spent within the 2m range of the visitors (which was our target for imitation), we would have needed to test approximately 7800 visitors to reach the same number of actual observation time (assuming a constant interaction time of the chimpanzees in close proximity). Testing such sample size would not have been feasible in the current study. Furthermore, it was not possible for us to know a priori the average interaction time (within the 2m range). Consequently, although we partially agree with the reviewer, we consider that the limitations pointed out could have hardly been addressed before data collection. Of course, we hope that future studies can expand on our results and account for potentially low interaction times between chimpanzees and zoo visitors.

To illustrate my issue more specifically: The authors state that the zoo guests (N = 50) participated for 5h30 in total, but as far as I can tell they do not directly state for how much of this time was a chimpanzee within 2m (i.e. their actual data collection window). I can only infer the true amount from the the following on Line 786.

Line 786: “The average duration of the visitors' engagement when potential imitative events took place was 14 min and 4 seconds (duration range 1 minute and 57 seconds

to 45 minutes and 12 seconds).”

—> **RESPONSE:** Following the reviewer suggestions we have included in line 827-829 the actual time that the chimpanzees and visitors stayed within 2m of each other (a total of 23 minutes and 46 seconds). Although we agree that a larger amount of time spent in close proximity between the chimpanzees and the visitors would have been desirable, this variable was outside our control. The chimpanzees at the testing institution have a large outdoor enclosure with several interconnected areas where they can freely move. Consequently, the degree of interaction with the visitors is decided by the chimpanzees themselves. As the outdoor enclosure of the chimpanzees at Leintal zoo is (most likely) larger than the indoor enclosure of the chimpanzees tested by Persson et al. (where inter-species interactions through a window took place), it is likely that the enclosure design had an effect on the time that the chimpanzees spent in close proximity of the zoo visitors (however please note that this is just a hypothesis as Persson et al. did not report enclosure sizes). We have added a discussion of this possibility in the Discussion in lines 996-1002.

Multiplying this average out by the number of participants for whom “potential imitative events” (chimpanzee approached within 2m) actually occurred (N = 9) would indicate that the authors only actually collected ~2h of actual data, from 9 human subjects and an unknown number of chimpanzees. This is, at best (i.e. not considering the fact that Persson et al. observed groups of humans), less than 5% of the observational data collected (52h) in the study the authors claim to be replicating.

—> **RESPONSE:** Please see our responses above explaining why it was not possible to estimate a priori the time the chimpanzees would choose to spend in close proximity of the zoo visitors and why the comparison with the "observational data" compiled by Persson et al. is not without limitation. We agree with the reviewer that we should have reported the total time spent within 2m between the chimpanzees and the visitors and we have added this information in line 827-829.

Line 934: Persson et al. (2017) only tested five chimpanzees, four of which had been at least partly human-reared. In our study, we tested 32 chimpanzees, only seven of which had been partly human-reared”

Given the issues described above, statements like this feel rather disingenuous. How many of these 32 chimpanzees actually came within 2m of the human participants (the author’s criteria for inclusion)? Given that only 30 potential imitative events occurred, it cannot be all of them. Any who did not do so were not “tested” and should be removed from this appeal to superior sample size. The authors did not observe these individuals.

—> **RESPONSE:** We respectfully disagree with the reviewer on this aspect. Persson et al. (2017) report that "The subjects were the five chimpanzees housed at Furuviik Zoo (Sweden)/Lund University Primate Research Station Furuviik in 2013" (p. 3) and that "Four of the five chimpanzees produced imitative actions, the exception being SF1, who hardly interacted with visitors" (p. 5). Consequently, Persson et al. (2017) considered that they tested all individuals included in the chimpanzee group, regardless if they imitated or if they interacted with the zoo visitors. Therefore, we similarly consider all chimpanzees included in the tested group as participants in our experiment (regardless if they interacted or not with

visitors). Following the reviewer's suggestion we specify in lines 852-856 which individuals interacted with the visitors during potential imitative events considered as such by both coders.

To be clear, chimpanzees approaching humans in close proximity is not in itself a behaviour of interest: The authors self-imposed a criteria of chimps being within 2m for ease of coding, not because imitative events can only occur, or are even necessarily most likely to occur, within this range. The term 'potential imitative event' is a poor fit then, since imitative events could occur at any number of other potential ranges, which the authors exclude by implication with this term.

—> **RESPONSE:** We agree with the reviewer that imitative events could also take place at longer distances than 2 meters. To avoid confusion we have made an edit in line 335-338 as follows: "During the Visitor experiment, the experimenter live-coded the following variables from each potential imitative event (for the purpose of this study, a potential imitative event was considered when the visitor was within 2 m of a chimpanzee through the glass window or the mesh of the outdoor enclosure)." We also added footnote number 4 (line 338) to clarify this point: "Note that imitative events can also take place at larger distances but for the purpose of this study the term "potential imitative events" will be used when the distance between a chimpanzee and a zoo visitor is less than 2m. This method was adopted based on the result of Persson et al. (2017) that 50% of the actions presumably imitated by the visitors and 72% of the actions presumably imitated by the chimpanzees occurred at a distance smaller than 2m"

Moreover, the fact that close-approaches were rare does not, therefore, present evidence either way for imitative behaviour in chimpanzees, although the authors make it sound so in the Discussion:

“We found that potential imitative events (when the chimpanzees and visitors were less than 2m apart through the glass window) were very rare. Even though we instructed the zoo visitors to try to make the chimpanzees imitate them (and consequently the visitors gestured and performed a wide variety of actions), the chimpanzees hardly ever came in close proximity with the visitors.”

—> **RESPONSE:** Although we agree with the reviewer that close-approaches at less than 2 m do not directly evidence presence or absence of imitation, we still believe that close-approaches (which we termed potential imitative events) are a useful proxy of the likelihood of inter-species imitation in zoo captive chimpanzees. If we look at the ethogram provided by Persson et al. (2017) of the imitative repertoires for visitors and chimpanzees (Table 2, p 7), we can observe that six of the behaviours described involve physical contact with the window separating the chimpanzees from the zoo visitors. Furthermore, these six behaviours account for 50% of the actions presumably imitated by the visitors and for 72% of the actions presumably imitated by the chimpanzees. Consequently, although it is true that close-proximity does not equate imitation, the majority of actions imitated by chimpanzees take place in very close proximity. It is also for this reason (and based on the results of Persson et al. 2017) that we originally considered potential imitative events as the interactions between zoo visitors and chimpanzees that took place less than 2m apart. We have now included this information in footnote number 4 (line 338).

This lab is known for its (well-justified!) claims that primatology has a bias towards methods

and mindsets which wrongly claim the existence of imitation and other ‘human-like’ behaviour in apes. I am sympathetic to this! However, it cuts both ways. It is hard to take seriously the disparity between the effort put into collecting the ‘behavioural baseline’ (forming the null hypothesis in this study (38 hours of observation, keeper surveys, etc) and the amount of actual data available to test it (2h of observation).

In light of this limitation, great caution should be taken in making any claims based on this data, and any inference drawn from them that Persson et al. “failed to replicate” or “does not generalise” should be removed from the manuscript. This was not a fair replication of their work.

—> **RESPONSE:** We are happy to hear that the reviewer shares our take on spontaneous ape imitation. Although we disagree with some of the criticism made by the reviewer, we agree that our data set was smaller than that of Persson et al. (2017) (although by how much is unclear given the limitations of their data collection method). Consequently we have been more cautious in the wording of our conclusions (see for example lines 977-980) and removed from the manuscript the expressions pointed out by the reviewer.

As a final methodological consideration: Is it possible that people being singly observed and recorded on video by researchers are more inhibited with regards to the actions they perform in front of the chimpanzees, either in terms of frequency or exaggeration of form? This seems likely to me, and places yet further importance on collecting a substantial amount of data to offset the effect.

—> **RESPONSE:** We agree with the reviewer that this is a possibility and we have now included this point in the discussion in lines 1007-1018.

Power Analysis

I commend the thorough power analysis conducted here, but I also question whether all 13 pages, 8 figures and 3 tables are conducive to comprehension and readability of the final manuscript (particularly since it does not refer to the actual sample used, see below). I suggest the authors at least consider breaking this up into further sub-sections that make it easier to follow, or (my preference) consider relegate some of this information or visualisations to supplemental material.

Line 967: “Using a simulation we estimated that in order to address our second aim with enough statistical power we needed to identify at least 46 imitation events. Although this number would need to be slightly adjusted due to the death of a chimpanzee included in our original estimation...”

—> **RESPONSE:** Following the reviewer's suggestion we have moved two figures (former Figures 1 and 2) and a table (former Table 4) of the power analysis to supplementary.

Why not just re-run the simulations using the actual sample size (N = 14, not 16) of chimpanzees? Maybe the authors are correct in thinking it won't make a difference, but we should not have to take their word on it.

—> **RESPONSE:** There are several reasons why we consider that it is unnecessary to re-

run the power analysis. First of all, the power analysis was conducted before data collected and included in the Stage 1 approved manuscript. Changing this analysis after collecting the data risks appearing disingenuous to the reader. Second, it is extremely unlikely that rerunning the power analysis changing the sample size from 16 to 14 would change our results. Even if the threshold of imitative events required to fit our model would change by 20 units (which is nearly impossible) we would still not have observed enough imitative events to fit our model with sufficient statistical power. Third, rerunning the power analysis would require a considerable time and CO₂ investment. Each time we had to run the simulation we required from 2 to 3 days due to the amount of computer power required. For the abovementioned reasons, we do not consider necessary or justified to rerun the simulation. However, if the reviewer or editor deem this issue a critical condition for publication, we will of course comply. We have added more information regarding why we refrained from re-running the simulation in lines 1038-1041 of the Discussion.

Demonstration experiment

No issues for me here, this seems a very robust experiment and the authors are even-handed about its outcomes. Well done!

—> **RESPONSE:** We thank the reviewer for this positive evaluation of our Demonstration experiment.

Minor comments and typological corrections

The abstract is rather vague. In present state, a reader knows a study has been replicated, but has no idea which one, nor are any of the current manuscript's methods described.

—> **RESPONSE:** Following the reviewer's comment we have rewritten parts of the abstract (lines 26-29) to add more information on our methodology and we have cited the paper we attempted to replicate (line 21-22).

Line 58: “demonstrated to the two chimpanzees 48 actions” should be “demosntrated 48 actions to the two chimpanzees”

—> **RESPONSE:** We have corrected this sentence as suggested by the reviewer in line 66.

Line 84: ‘Contextual imitation’ needs to be more clearly defined on its first usage.

—> **RESPONSE:** We have reorganized this sentence to clarify the concept of contextual imitation in line 93.

Line 136: ‘et al’ needs a full stop after ‘al’

—> **RESPONSE:** We have now corrected this typo in line 128

Line 143-148: How does this fourth thing differ from the ‘baseline repertoire’ referenced as the first thing on line 136?

—> **RESPONSE:** Both limitations are related. The fact the Persson et al. considered as imitated actions behaviours that had not been seen in the last 3 minutes is an arbitrary and likely incorrect measure. By using the cut off point of 3 minutes the experimenters conflated contextual imitation (performance of a behaviour that is already in the individual's repertoire) with production imitation (performance of a novel behaviour acquired from the observation of a demonstrator).

Line 183: It's not clear here whether the power analysis is relevant to both observation and demonstration experiments, or just the latter. It is also not clear what the 'average' being referred to is – averaged across what?

—> **RESPONSE:** For the sake of clarity we have removed "average" from this sentence and we have added that the power analysis refers to the Demonstration experiment in line 194.

in front of the chimpanzees, either in terms of frequency or exaggeration of form? This seems likely to me, and places yet further importance on collecting a substantial amount of data to offset the effect.

—> **RESPONSE: We agree with the reviewer that this is a possibility and we have now included this point in the discussion in lines 1007-1018.**

Power Analysis

I commend the thorough power analysis conducted here, but I also question whether all 13 pages, 8 figures and 3 tables are conducive to comprehension and readability of the final manuscript (particularly since it does not refer to the actual sample used, see below). I suggest the authors at least consider breaking this up into further sub-sections that make it easier to follow, or (my preference) consider relegate some of this information or visualisations to supplemental material.

Line 967: “Using a simulation we estimated that in order to address our second aim with enough statistical power we needed to identify at least 46 imitation events. Although this number would need to be slightly adjusted due to the death of a chimpanzee included in our original estimation...”

—> **RESPONSE: Following the reviewer's suggestion we have moved two figures (former Figures 1 and 2) and a table (former Table 4) of the power analysis to supplementary.**

Why not just re-run the simulations using the actual sample size (N = 14, not 16) of chimpanzees? Maybe the authors are correct in thinking it is won't make a difference, but we should not have to take their word on it.

—> **RESPONSE: There are several reasons why we consider that it is unnecessary to re-run the power analysis. First of all, the power analysis was conducted before data collected and included in the Stage 1 approved manuscript. Changing this analysis after collecting the data could seem disingenuous to the reader. Second, it is extremely unlikely that rerunning the power analysis changing the sample size from 16 to 14 would change our results. Even if the threshold of imitative events required to fit our model would change by 20 units (which is nearly impossible) we would still not have observed enough imitative events to fit our model with sufficient statistical power. Third, rerunning the power analysis would require a considerable time and CO₂ investment. Each time we had to run the simulation we required from 2 to 3 days due to the amount of computer power required. For the abovementioned reasons, we do not consider necessary to rerun the simulation. However, if the reviewer or editor deem this issue a critical condition for publication, we will of course comply. We have added more information regarding why we refrained from re-running the simulation in lines 1038-1041 of the Discussion.**

Demonstration experiment

No issues for me here, this seems a very robust experiment and the authors are even-handed

about its outcomes. Well done!

—> **RESPONSE:** We thank the reviewer for this positive evaluation of our experiment.

Minor comments and typological corrections

The abstract is rather vague. In present state, a reader knows a study has been replicated, but has no idea which one, nor are any of the current manuscript's methods described.

—> **RESPONSE:** Following the reviewer's comment we have rewritten parts of the abstract (lines 26-29) to add more information on our methodology and we have cited the paper we attempted to replicate (line 21-22).

Line 58: “demonstrated to the two chimpanzees 48 actions” should be “demonstrated 48 actions to the two chimpanzees”

—> **RESPONSE:** We have corrected this sentence as suggested by the reviewer in line 66.

Line 84: ‘Contextual imitation’ needs to be more clearly defined on its first usage.

—> **RESPONSE:** We have reorganized this sentence to clarify the concept of contextual imitation in line 93.

Line 136: ‘et al’ needs a full stop after ‘al’

—> **RESPONSE:** We have now corrected this typo in line 128

Line 143-148: How does this fourth thing differ from the ‘baseline repertoire’ referenced as the first thing on line 136?

—> **RESPONSE:** Both limitations are related. The fact the Persson et al. considered as imitated actions behaviours that had not been seen in the last 3 minutes is an arbitrary and incorrect measure. By using the cut off point of 3 minutes the experimenters conflated contextual imitation (performance of a behaviour that is already in the individual's repertoire) with production imitation (performance of a novel behaviour acquired from the observation of a demonstrator).

Line 183: It's not clear here whether the power analysis is relevant to both observation and demonstration experiments, or just the latter. It is also not clear what the ‘average’ being referred to is – averaged across what?

—> **RESPONSE:** For the sake of clarity we have removed "average" from this sentence and we have added that the power analysis refers to the Demonstration experiment in line 194.